# Fast Partitioned Learned Bloom Filter

**Atsuki Sato**    **Yusuke Matsui**
The University of Tokyo
Tokyo, Japan
a_sato@hal.t.u-tokyo.ac.jp    matsui@hal.t.u-tokyo.ac.jp

## Abstract

A Bloom filter is a memory-efficient data structure for approximate membership queries used in numerous fields of computer science. Recently, learned Bloom filters that achieve better memory efficiency using machine learning models have attracted attention. One such filter, the partitioned learned Bloom filter (PLBF), achieves excellent memory efficiency. However, PLBF requires a $\mathcal{O}(N^3k)$ time complexity to construct the data structure, where $N$ and $k$ are the hyperparameters of PLBF. One can improve memory efficiency by increasing $N$, but the construction time becomes extremely long. Thus, we propose two methods that can reduce the construction time while maintaining the memory efficiency of PLBF. First, we propose fast PLBF, which can construct the same data structure as PLBF with a smaller time complexity $\mathcal{O}(N^2k)$. Second, we propose fast PLBF++, which can construct the data structure with even smaller time complexity $\mathcal{O}(Nk \log N + Nk^2)$. Fast PLBF++ does not necessarily construct the same data structure as PLBF. Still, it is almost as memory efficient as PLBF, and it is proved that fast PLBF++ has the same data structure as PLBF when the distribution satisfies a certain constraint. Our experimental results from real-world datasets show that (i) fast PLBF and fast PLBF++ can construct the data structure up to 233 and 761 times faster than PLBF, (ii) fast PLBF can achieve the same memory efficiency as PLBF, and (iii) fast PLBF++ can achieve almost the same memory efficiency as PLBF. The codes are available at https://github.com/atsukisato/FastPLBF.

## 1 Introduction

Membership query is a problem of determining whether a given query $q$ is contained within a set $\mathcal{S}$. Membership query is widely used in numerous areas, including networks and databases. One can correctly answer the membership query by keeping the set $\mathcal{S}$ and checking whether it contains $q$. However, this approach is memory intensive because we need to maintain $\mathcal{S}$.

A Bloom filter [1] is a memory-efficient data structure that answers approximate membership queries. A Bloom filter uses hash functions to compress the set into a bitstring and then answers the queries using the bitstring. When a Bloom filter answers $q \in \mathcal{S}$, it can be wrong (false positives can arise); when it answers $q \notin \mathcal{S}$, it is always correct (no false negatives arise). A Bloom filter has been incorporated into numerous systems, such as networks [2, 3, 4], databases [5, 6, 7], and cryptocurrencies [8, 9].

A traditional Bloom filter does not care about the distribution of the set or the queries. Therefore, even if the distribution has a characteristic structure, a Bloom filter cannot take advantage of it. Kraska et al. proposed learned Bloom filters (LBFs), which utilize the distribution structure using a machine learning model [10]. LBF achieves a better trade-off between memory usage and false positive rate (FPR) than the original Bloom filter. Since then, several studies have been conducted on various LBFs [11, 12, 13].

37th Conference on Neural Information Processing Systems (NeurIPS 2023).

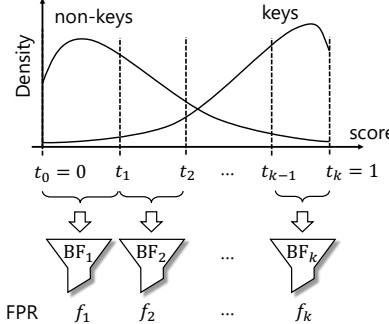

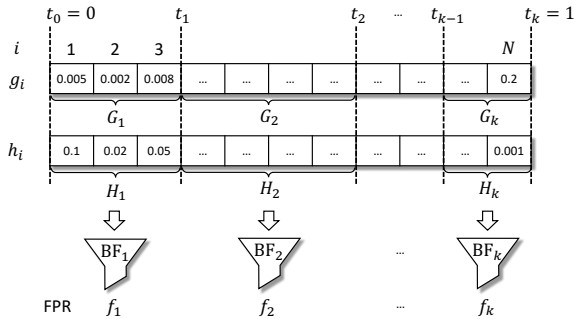

Figure 1: PLBF partitions the score space into $k$ regions and assigns backup Bloom filters with different FPRs to each region.

Figure 2: PLBF divides the score space into $N$ segments and then clusters the $N$ segments into $k$ regions. PLBF uses dynamic programming to find the optimal way to cluster segments into regions.

Partitioned learned Bloom filter (PLBF) [14] is a variant of the LBF. PLBF effectively uses the distribution and is currently one of the most memory-efficient LBFs. To construct PLBF, a machine learning model is trained to predict whether the input is included in the set. For a given element $x$, the machine learning model outputs a score $s(x) \in [0, 1]$, indicating the probability that $\mathcal{S}$ includes $x$. PLBF divides the score space $[0, 1]$ into $N$ equal segments and then appropriately clusters the $N$ segments into $k(< N)$ regions using dynamic programming (DP). Then, $k$ backup Bloom filters with different FPRs are assigned to each region. We can obtain a better (closer to optimal) data structure with a larger $N$. However, PLBF builds DP tables $\mathcal{O}(N)$ times, and each DP table requires $\mathcal{O}(N^2 k)$ time complexity to build, so the total time complexity amounts to $\mathcal{O}(N^3 k)$. Therefore, the construction time increases rapidly as $N$ increases.

We propose fast PLBF, which is constructed much faster than PLBF. Fast PLBF constructs the same data structure as PLBF but with $\mathcal{O}(N^2 k)$ time complexity by omitting the redundant construction of DP tables. Furthermore, we propose fast PLBF++, which can be constructed faster than fast PLBF, with a time complexity of $\mathcal{O}(Nk \log N + Nk^2)$. Fast PLBF++ accelerates DP table construction by taking advantage of a characteristic that DP tables often have. We proved that fast PLBF++ constructs the same data structure as PLBF when the probability of $x \in \mathcal{S}$ and the score $s(x)$ are ideally well correlated. Our contributions can be summarized as follows.

- We propose fast PLBF, which can construct the same data structure as PLBF with less time complexity.
- We propose fast PLBF++, which can construct the data structure with even less time complexity than fast PLBF. Fast PLBF++ constructs the same data structure as PLBF when the probability of $x \in \mathcal{S}$ and the score $s(x)$ are ideally well correlated.
- Experimental results show that fast PLBF can construct the data structure up to 233 times faster than PLBF and achieves the same memory efficiency as PLBF.
- Experimental results show that fast PLBF++ can construct the data structure up to 761 times faster than PLBF and achieves almost the same memory efficiency as PLBF.

## 2 Preliminaries: PLBF

First, we define terms to describe PLBF [14]. Let $\mathcal{S}$ be a set of elements for which the Bloom filter is to be built, and let $\mathcal{Q}$ be the set of elements not included in $\mathcal{S}$ that is used when constructing PLBF ($\mathcal{S} \cap \mathcal{Q} = \varnothing$). The elements included in $\mathcal{S}$ are called keys, and those not included are called non-keys. To build PLBF, a machine learning model is trained to predict whether a given element $x$ is included in the set $\mathcal{S}$ or $\mathcal{Q}$. For a given element $x$, the machine learning model outputs a score $s(x) \in [0, 1]$. The score $s(x)$ indicates "how likely is $x$ to be included in the set $\mathcal{S}$."

Next, we explain the design of PLBF. PLBF partitions the score space $[0, 1]$ into $k$ regions and assigns backup Bloom filters with different FPRs to each region (Figure 1). Given a target overall FPR,

$F \in (0, 1)$, we optimize $\boldsymbol{t} \in \mathbb{R}^{k+1}$ and $\boldsymbol{f} \in \mathbb{R}^k$ to minimize the total memory usage. Here, $\boldsymbol{t}$ is a vector of thresholds for partitioning the score space into $k$ regions, and $\boldsymbol{f}$ is a vector of FPRs for each region, satisfying $t_0 = 0 < t_1 < \cdots < t_k = 1$ and $0 < f_i \le 1$ $(i = 1 \dots k)$.

Next, we explain how PLBF finds the optimal $\boldsymbol{t}$ and $\boldsymbol{f}$. PLBF divides the score space $[0, 1]$ into $N(> k)$ segments and then finds the optimal $\boldsymbol{t}$ and $\boldsymbol{f}$ using DP. Deciding how to cluster $N$ segments into $k$ consecutive regions corresponds to determining the threshold $\boldsymbol{t}$ (Figure 2). We denote the probabilities that the key and non-key scores are contained in the $i$-th **segment** by $g_i$ and $h_i$, respectively. After we cluster the segments (i.e., determine the thresholds $\boldsymbol{t}$), we denote the probabilities that the key and non-key scores are contained in the $i$-th **region** by $G_i$ and $H_i$, respectively (e.g., $g_1 + g_2 + g_3 = G_1$ in Figure 2).

We can find the thresholds $\boldsymbol{t}$ that minimize memory usage by solving the following problem for each $j = k \dots N$ (see the appendix for details); we find a way to cluster the 1st to $(j - 1)$-th segments into $k - 1$ regions while maximizing

$$\sum_{i=1}^{k-1} G_i \log_2 \left( \frac{G_i}{H_i} \right). \tag{1}$$

PLBF solves this problem by building a $j \times k$ DP table $\mathrm{DP}_{\mathrm{KL}}^j[p][q]$ $(p = 0 \dots j-1$ and $q = 0 \dots k-1)$ for each $j$. $\mathrm{DP}_{\mathrm{KL}}^j[p][q]$ denotes the maximum value of $\sum_{i=1}^q G_i \log_2 \left( \frac{G_i}{H_i} \right)$ one can get when you cluster the 1st to $p$-th segments into $q$ regions. To construct PLBF, one must find a clustering method that achieves $\mathrm{DP}_{\mathrm{KL}}^j[j - 1][k - 1]$. $\mathrm{DP}_{\mathrm{KL}}^j$ can be computed recursively as follows:

$$\mathrm{DP}_{\mathrm{KL}}^j[p][q] = \begin{cases} 0 & (p = 0 \wedge q = 0) \\ -\infty & ((p = 0 \wedge q > 0) \vee (p > 0 \wedge q = 0)) \\ \max_{i=1 \dots p} \left( \mathrm{DP}_{\mathrm{KL}}^j[i - 1][q - 1] + d_{\mathrm{KL}}(i, p) \right) & (\text{else}), \end{cases} \tag{2}$$

where the function $d_{\mathrm{KL}}(i_l, i_r)$ is the following function defined for integers $i_l$ and $i_r$ satisfying $1 \le i_l \le i_r \le N$:

$$d_{\mathrm{KL}}(i_l, i_r) = \left( \sum_{i=i_l}^{i_r} g_i \right) \log_2 \left( \frac{\sum_{i=i_l}^{i_r} g_i}{\sum_{i=i_l}^{i_r} h_i} \right). \tag{3}$$

The time complexity to construct this DP table is $\mathcal{O}(j^2 k)$. Then, by tracing the recorded transitions backward from $\mathrm{DP}_{\mathrm{KL}}^j[j - 1][k - 1]$, we obtain the best clustering with a time complexity of $\mathcal{O}(k)$. As the DP table is constructed for each $j = k \dots N$, the overall complexity is $\mathcal{O}(N^3 k)$. The pseudo-code for PLBF construction is provided in the appendix.

We can divide the score space more finely with a larger $N$ and thus obtain a near-optimal $\boldsymbol{t}$. However, the time complexity increases rapidly with increasing $N$.

## 3 Fast PLBF

We propose fast PLBF, which constructs the same data structure as PLBF more quickly than PLBF by omitting the redundant construction of DP tables. Fast PLBF uses the same design as PLBF and finds the best clustering (i.e., $\boldsymbol{t}$) and FPRs (i.e., $\boldsymbol{f}$) to minimize memory usage.

PLBF constructs a DP table for each $j = k \dots N$. We found that this computation is redundant and that we can also use the last DP table $\mathrm{DP}_{\mathrm{KL}}^N$ for $j = k \dots N - 1$. This is because the maximum value of $\sum_{i=1}^{k-1} G_i \log_2 \left( \frac{G_i}{H_i} \right)$ when clustering the 1st to $(j - 1)$-th segments into $k - 1$ regions is equal to $\mathrm{DP}_{\mathrm{KL}}^N[j - 1][k - 1]$. We can obtain the best clustering by tracing the transitions backward from $\mathrm{DP}_{\mathrm{KL}}^N[j - 1][k - 1]$. The time complexity of tracing the transitions is $\mathcal{O}(k)$, which is faster than constructing the DP table.

This is a simple method, but it is a method that only becomes apparent after some organization on the optimization problem of PLBF. PLBF solves $N - k + 1$ optimization problems separately, i.e., for each of $j = k, \dots, N$, the problem of "finding the optimal $\boldsymbol{t}$ and $\boldsymbol{f}$ when the $k$-th region consists of

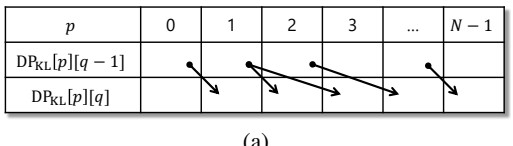 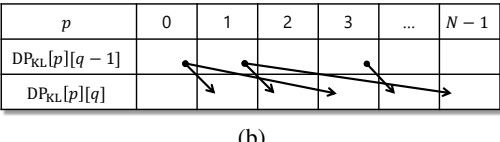

(a)                                  (b)

Figure 3: The cases (a) where there is no "crossing" of the recorded transitions when computing $\mathrm{DP}_{\mathrm{KL}}[p][q]$ ($p = 1 \ldots N - 1$) from $\mathrm{DP}_{\mathrm{KL}}[p][q - 1]$ ($p = 0 \ldots N - 2$) and (b) where there is. The arrows indicate which transition took the maximum value. Empirically, the probability of "crossing" is small.

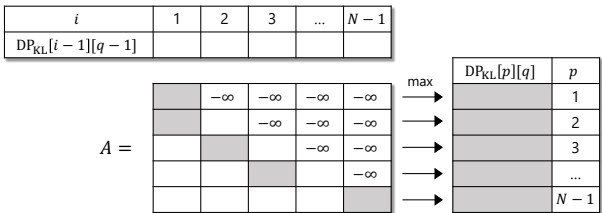 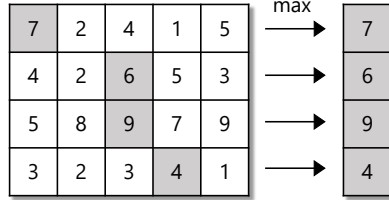

Figure 4: Computing $\mathrm{DP}_{\mathrm{KL}}[p][q]$ ($p = 1 \ldots N - 1$) from $\mathrm{DP}_{\mathrm{KL}}[p][q - 1]$ ($p = 0 \ldots N - 2$) via the matrix $A$. The computation is the same as solving the matrix problem for matrix $A$. When the score distribution is *ideal*, $A$ is a *monotone matrix*.

Figure 5: Example of a *matrix problem* for a $4 \times 5$ *monotone matrix*. The position of the maximum value in each row (filled in gray) is moved to the "lower right".

the $j, \ldots, N$th segments" is solved separately. Fast PLBF, on the other hand, solves the problems for $j = k, \ldots, N - 1$ faster by reusing the computations in the problem for $j = N$. The reorganization of the problem in the appendix makes it clear that this reuse does not change the answer.

The pseudo-code for fast PLBF construction is provided in the appendix. The time complexity of building $\mathrm{DP}_{\mathrm{KL}}^N$ is $\mathcal{O}(N^2 k)$, and the worst-case complexity of subsequent computations is $\mathcal{O}(Nk^2)$. Because $N > k$, the total complexity is $\mathcal{O}(N^2 k)$, which is faster than $\mathcal{O}(N^3 k)$ for PLBF, although fast PLBF constructs the same data structure as PLBF.

Fast PLBF extends the usability of PLBF. In any application of PLBF, fast PLBF can be used instead of PLBF because fast PLBF can be constructed quickly without losing the accuracy of PLBF. Fast PLBF has applications in a wide range of computing areas [15, 5], and is significantly superior in applications where construction is frequently performed. Fast PLBF also has the advantage of simplifying hyperparameter settings. When using PLBF, the hyperparameters $N$ and $k$ must be carefully determined, considering the trade-off between accuracy and construction speed. With fast PLBF, on the other hand, it is easy to determine the appropriate hyperparameters because the construction time is short enough, even if $N$ and $k$ are set somewhat large.

## 4 Fast PLBF++

We propose fast PLBF++, which can be constructed even faster than fast PLBF. Fast PLBF++ accelerates the construction of the DP table $\mathrm{DP}_{\mathrm{KL}}^N$ by taking advantage of a characteristic that DP tables often have. When the transitions recorded in computing $\mathrm{DP}_{\mathrm{KL}}^N[p][q]$ ($p = 1 \ldots N - 1$) from $\mathrm{DP}_{\mathrm{KL}}^N[p][q - 1]$ ($p = 0 \ldots N - 2$) are represented by arrows, as in Figure 3, we find that the arrows rarely "cross" (at locations other than endpoints). In other words, the transitions tend to have few intersections (Figure 3(a)) rather than many (Figure 3(b)). Fast PLBF++ takes advantage of this characteristic to construct the DP table with less complexity $\mathcal{O}(Nk \log N)$, thereby reducing the total construction complexity to $\mathcal{O}(Nk \log N + Nk^2)$.

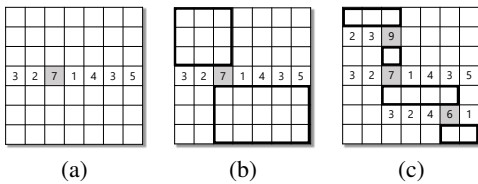

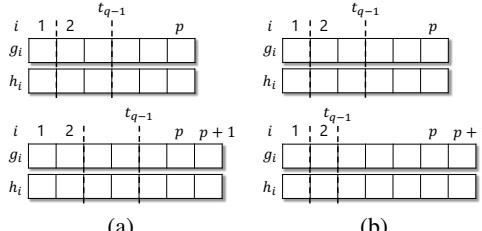

Figure 6: A divide-and-conquer algorithm for solving a *matrix problem* for a *monotone matrix*: (a) An exhaustive search is performed on the middle row, and (b) the result is used to narrow down the search area. (c) This is repeated recursively.

Figure 7: When the number of regions is fixed at $q$ and the number of segments is increased by 1, the optimal $t_{q-1}$ remains unchanged or increases if $A$ is a *monotone matrix* (a). When $A$ is not a *monotone matrix* (b), the optimal $t_{q-1}$ may decrease.

First, we define the terms to describe fast PLBF++. For simplicity, $\text{DP}_{\text{KL}}^N$ is denoted as $\text{DP}_{\text{KL}}$ in this section. The $(N-1) \times (N-1)$ matrix $A$ is defined as follows:

$$A_{pi} = \begin{cases} -\infty & (i = p+1, p+2, \ldots, N-1) \\ \text{DP}_{\text{KL}}[i-1][q-1] + d_{\text{KL}}(i, p) & (\text{else}). \end{cases} \tag{4}$$

Then, from the definition of $\text{DP}_{\text{KL}}$,

$$\text{DP}_{\text{KL}}[p][q] = \max_{i=1\ldots N-1} A_{pi}. \tag{5}$$

The matrix $A$ represents the intermediate calculations involved in determining $\text{DP}_{\text{KL}}[p][q]$ from $\text{DP}_{\text{KL}}[p][q-1]$ (Figure 4).

Following Aggarwal et al. [16], we define the *monotone matrix* and *matrix problem* as follows.

**Definition 4.1.** Let $B$ be an $n \times m$ real matrix, and we define a function $J : \{1 \ldots n\} \to \{1 \ldots m\}$, where $J(i)$ is the $j \in \{1 \ldots m\}$ such that $B_{ij}$ is the maximum value of the $i$-th row of $B$. If there is more than one such $j$, let $J(i)$ be the smallest. A matrix $B$ is called a *monotone matrix* if $J(i_1) \leq J(i_2)$ for any $i_1$ and $i_2$ that satisfy $1 \leq i_1 < i_2 \leq n$. Finding the maximum value of each row of a matrix is called a *matrix problem*.

An example of a *matrix problem* for a *monotone matrix* is shown in Figure 5. Solving the *matrix problem* for a general $n \times m$ matrix requires $\mathcal{O}(nm)$ time complexity because all matrix values must be checked. Meanwhile, if the matrix is known to be a *monotone matrix*, the *matrix problem* for this matrix can be solved with a time complexity of $\mathcal{O}(n + m \log n)$ using the divide-and-conquer algorithm [16]. Here, the exhaustive search for the middle row and the refinement of the search range are repeated recursively (Figure 6).

We also define an *ideal score distribution* as follows.

**Definition 4.2.** A score distribution is *ideal* if the following holds:

$$\frac{g_1}{h_1} \leq \frac{g_2}{h_2} \leq \cdots \leq \frac{g_N}{h_N}. \tag{6}$$

An *ideal score distribution* implies that the probability of $x \in \mathcal{S}$ and the score $s(x)$ are ideally well correlated. In other words, an *ideal score distribution* means that the machine learning model learns the distribution ideally.

"Few crossing transitions" in Figure 3 indicates that $A$ is a *monotone matrix* or is close to it. It is somewhat intuitive that $A$ is a *monotone matrix* or is close to it. This is because the fact that $A$ is a *monotone matrix* implies that the optimal $t_{q-1}$ does not decrease when the number of regions is fixed at $q$ and the number of segments increases by 1 (Figure 7). It is intuitively more likely that $t_{q-1}$ remains unchanged or increases, as in Figure 7(a), than that $t_{q-1}$ decreases, as in Figure 7(b). Fast PLBF++ takes advantage of this insight to rapidly construct DP tables.

From Equation (5), determining $\text{DP}_{\text{KL}}[p][q]$ $(p = 1 \ldots N-1)$ from $\text{DP}_{\text{KL}}[p][q-1]$ $(p = 0 \ldots N-2)$ is equivalent to solving the *matrix problem* of matrix $A$. When $A$ is a *monotone matrix*, the divide-and-conquer algorithm can solve this problem with $\mathcal{O}(N \log N)$ time complexity. (The same algorithm

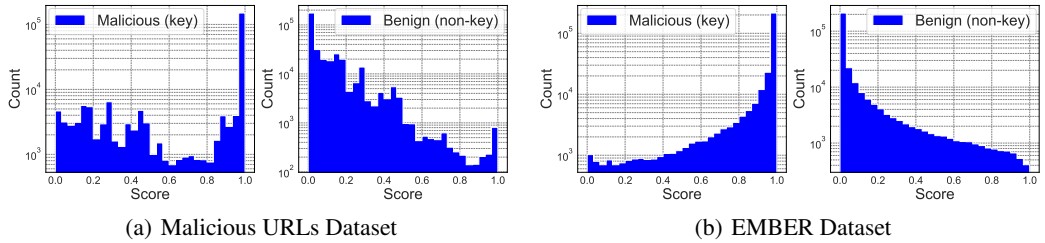

(a) Malicious URLs Dataset          (b) EMBER Dataset

Figure 8: Histograms of the score distributions of keys and non-keys.

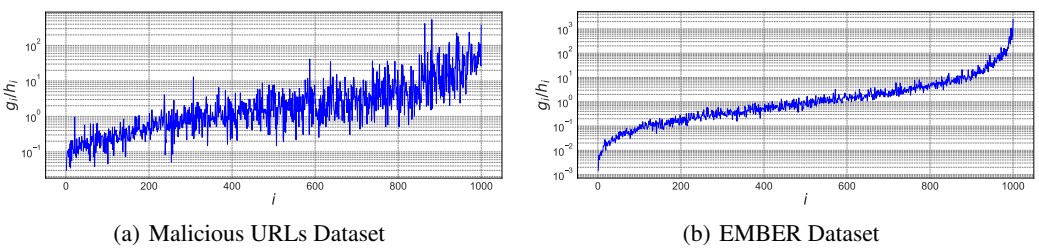

(a) Malicious URLs Dataset          (b) EMBER Dataset

Figure 9: Ratio of keys to non-keys.

can obtain a not necessarily correct solution even if $A$ is not a *monotone matrix*.) By computing $\mathrm{DP_{KL}}[p][q]$ with this algorithm sequentially for $q = 1 \ldots k-1$, we can construct a DP table with a time complexity of $\mathcal{O}(Nk \log N)$. The worst-case complexity of the subsequent calculations is $\mathcal{O}(Nk^2)$, so the total complexity is $\mathcal{O}(Nk \log N + Nk^2)$. This is the fast PLBF++ construction algorithm, and this is faster than fast PLBF, which requires $\mathcal{O}(N^2k)$ computations.

Fast PLBF++ does not necessarily have the same data structure as PLBF because $A$ is not necessarily a *monotone matrix*. However, as the following theorem shows, we can prove that $A$ is a *monotone matrix* under certain conditions.

**Theorem 4.3.** *If the score distribution is ideal, $A$ is a monotone matrix.*

The proof is given in the appendix. When the distribution is not *ideal*, matrix $A$ is not necessarily a *monotone matrix*, but as mentioned above, it is somewhat intuitive that $A$ is close to a *monotone matrix*. In addition, as will be shown in the next section, experiment results from real-world datasets whose distribution is not *ideal* show that fast PLBF++ is almost as memory efficient as PLBF.

## 5 Experiments

This section evaluates the experimental performance of fast PLBF and fast PLBF++. We compared the performances of fast PLBF and fast PLBF++ with four baselines: Bloom filter [1], Ada-BF [11], sandwiched learned Bloom filter (sandwiched LBF) [12], and PLBF [14]. Similar to PLBF, Ada-BF is an LBF that partitions the score space into several regions and assigns different FPRs to each region. However, Ada-BF relies heavily on heuristics for clustering and assigning FPRs. Sandwiched LBF is an LBF that "sandwiches" a machine learning model with two Bloom filters. This achieves better memory efficiency than the original LBF by optimizing the size of two Bloom filters.

To facilitate the comparison of different methods or hyperparameters results, we have slightly modified the original PLBF framework. The original PLBF was designed to minimize memory usage under the condition of a given false positive rate. However, this approach makes it difficult to compare the results of different methods or hyperparameters. This is because both the false positive rate at test time and the memory usage vary depending on the method and hyperparameters, which often makes it difficult to determine the superiority of the results. Therefore, in our experiments, we used a framework where the expected false positive rate is minimized under the condition of memory usage. This approach makes it easy to obtain two results with the same memory usage and compare them by

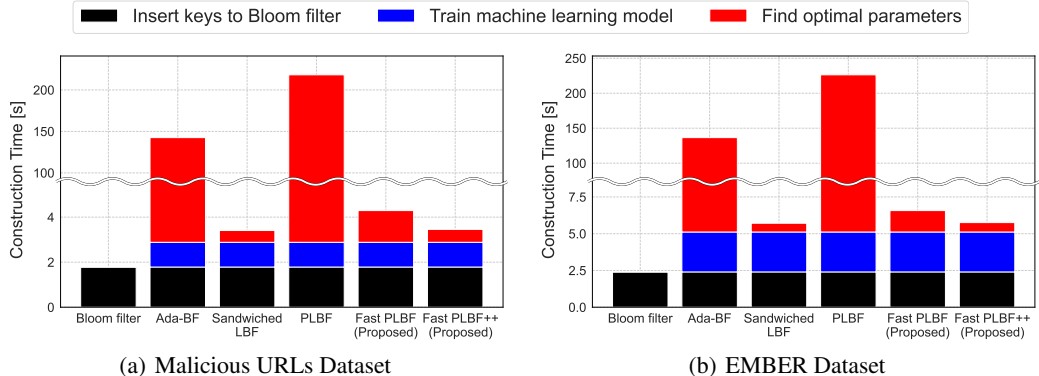

(a) Malicious URLs Dataset  (b) EMBER Dataset

Figure 10: Construction time.

the false positive rate at test time. See the appendix for more information on how this framework modification will change the construction method of PLBFs.

**Datasets**: We evaluated the algorithms using the following two datasets.

- **Malicious URLs Dataset**: As in previous papers [11, 14], we used Malicious URLs Dataset [17]. The URLs dataset comprises 223,088 malicious and 428,118 benign URLs. We extracted 20 lexical features such as URL length, use of shortening, number of special characters, etc. We used all malicious URLs and 342,482 (80%) benign URLs as the training set, and the remaining benign URLs as the test set.
- **EMBER Dataset**: We used the EMBER dataset [18] as in the PLBF research. The dataset consists of 300,000 malicious and 400,000 benign files, along with the features of each file. We used all malicious files and 300,000 (75%) benign files as the train set and the remaining benign files as the test set.

While any model can be used for the classifier, we used LightGBM [19] because of its speed in training and inference, as well as its memory efficiency and accuracy. The sizes of the machine learning model for the URLs and EMBER datasets are 312 Kb and 1.19 Mb, respectively. The training time of the machine learning model for the URLs and EMBER datasets is 1.09 and 2.71 seconds, respectively. The memory usage of LBF is the total memory usage of the backup Bloom filters and the machine learning model. Figure 8 shows a histogram of each dataset's score distributions of keys and non-keys. We can see that the frequency of keys increases and that of non-keys decreases as the score increases. In addition, Figure 9 plots $g_i/h_i$ $(i = 1 \ldots N)$ when $N = 1,000$. We can see that $g_i/h_i$ tends to increase as $i$ increases, but the increase is not monotonic (i.e., the score distribution is not *ideal*).

## 5.1 Construction time

We compared the construction times of fast PLBF and fast PLBF++ with those of existing methods. Following the experiments in the PLBF paper, hyperparameters for PLBF, fast PLBF, and fast PLBF++ were set to $N = 1,000$ and $k = 5$.

Figure 10 shows the construction time for each method. The construction time for learned Bloom filters includes not only the time to insert keys into the Bloom filters but also the time to train the machine learning model and the time to compute the optimal parameters ($t$ and $f$ in the case of PLBF).

Ada-BF and sandwiched LBF use heuristics to find the optimal parameters, so they have shorter construction times than PLBF but have worse accuracy. PLBF has better accuracy but takes more than 3 minutes to find the optimal $t$ and $f$. On the other hand, our fast PLBF and fast PLBF++ take less than 2 seconds. As a result, Fast PLBF constructs 50.8 and 34.3 times faster than PLBF, and fast PLBF++ constructs 63.1 and 39.3 times faster than PLBF for the URLs and EMBER datasets, respectively. This is about the same construction time as sandwiched LBF, which relies heavily on heuristics and, as we will see in the next section, is much less accurate than PLBF.

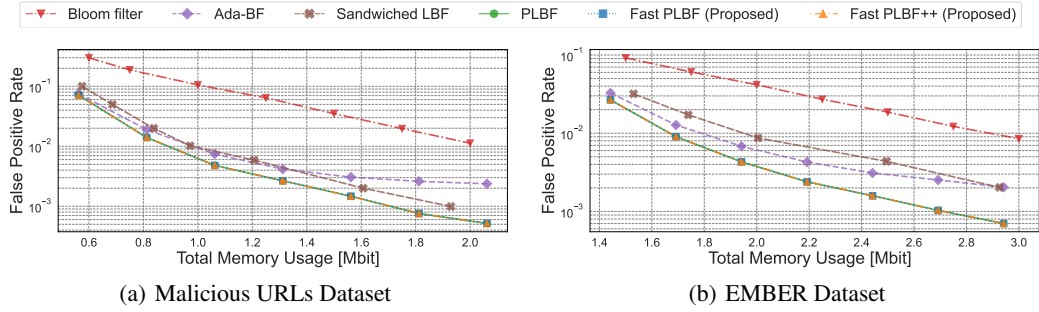

(a) Malicious URLs Dataset

(b) EMBER Dataset

Figure 11: Trade-off between memory usage and FPR.

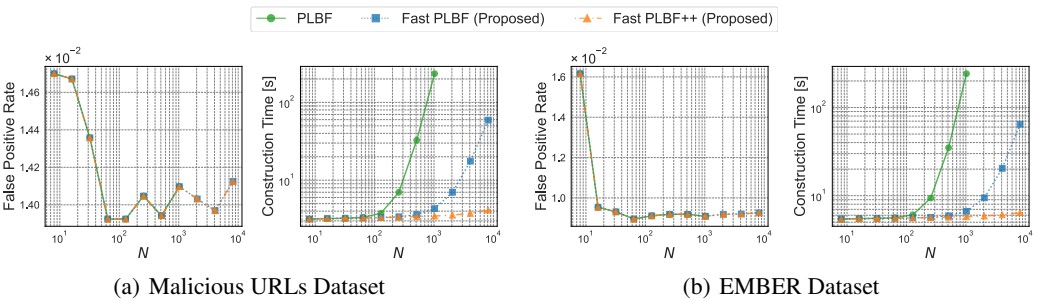

(a) Malicious URLs Dataset

(b) EMBER Dataset

Figure 12: Ablation study for hyper-parameter $N$.

## 5.2 Memory Usage and FPR

We compared the trade-off between memory usage and FPR for fast PLBF and fast PLBF++ with Bloom filter, Ada-BF, sandwiched LBF, and PLBF. Following the experiments in the PLBF paper, hyperparameters for PLBF, fast PLBF, and fast PLBF++ were always set to $N = 1,000$ and $k = 5$.

Figure 11 shows each method's trade-off between memory usage and FPR. PLBF, fast PLBF, and fast PLBF++ have better Pareto curves than the other methods for all datasets. Fast PLBF constructs the same data structure as PLBF in all cases, so it has exactly the same accuracy as PLBF. Fast PLBF++ achieves almost the same accuracy as PLBF. Fast PLBF++ has up to 1.0019 and 1.000083 times higher false positive rates than PLBF for the URLs and EMBER datasets, respectively.

## 5.3 Ablation study for hyper-parameters

The parameters of the PLBFs are memory size, $N$, and $k$. The memory size is specified by the user, and $N$ and $k$ are hyperparameters that are determined by balancing construction time and accuracy. In the previous sections, we set $N$ to 1,000 and $k$ to 5, following the original paper on PLBF. In this section, we perform ablation studies for these hyperparameters to confirm that our proposed methods can construct accurate data structures quickly, no matter what hyperparameter settings are used. We also confirm that the accuracy tends to be better and the construction time increases as $N$ and $k$ are increased, and that the construction time of the proposed methods increases much slower than PLBF.

Figure 12 shows the construction time and false positive rate with various $N$ while the memory usage of the backup Bloom filters is fixed at 500 Kb and $k$ is fixed at 5. For all three PLBFs, the false positive rate tends to decrease as $N$ increases. (Note that this is the false positive rate on test data, so it does not necessarily decrease monotonically. The appendix shows that the expected value of the false positive rate calculated using training data decreases monotonically as $N$ increases.) Also, as $N$ increases, the PLBF construction time increases rapidly, but the fast PLBF construction time increases much more slowly than that, and for fast PLBF++, the construction time changes little. This is because the construction time of PLBF is asymptotically proportional to $N^3$, while that of fast PLBF and fast PLBF++ is proportional to $N^2$ and $N \log N$, respectively. The experimental results

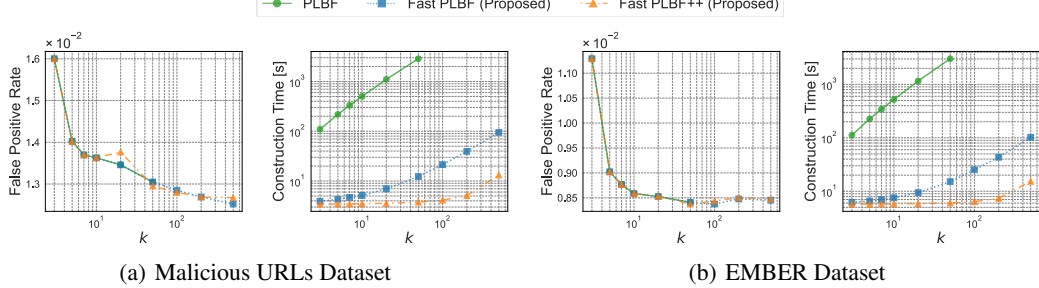

Figure 13: Ablation study for hyper-parameter $k$.

show that the two proposed methods can achieve high accuracy without significantly changing the construction time with large $N$.

Figure 13 shows the construction time and false positive rate with various $k$ while the backup bloom filter memory usage is fixed at 500 Kb and $N$ is fixed at 1,000. For all three PLBFs, the false positive rate tends to decrease as $k$ increases. For the EMBER dataset, the false positive rate stops decreasing at about $k = 20$, while for the URLs dataset, it continues to decrease even at about $k = 500$. (Just as in the case of experiments with varying $N$, this decrease is not necessarily monotonic.) In addition, the construction times of all three PLBFs increase proportionally to $k$, but fast PLBF has a much shorter construction time than PLBF, and fast PLBF++ has an even shorter construction time than fast PLBF. When $k = 50$, fast PLBF constructs 233 and 199 times faster than PLBF, and fast PLBF++ constructs 761 and 500 times faster than PLBF for the URLs and EMBER datasets, respectively. The experimental results indicate that by increasing $k$, the two proposed methods can achieve high accuracy without significantly affecting the construction time.

# 6   Related Work

Approximate membership query is a query that asks whether the query $q$ is contained in the set $\mathcal{S}$ while allowing for false positives with a small probability $\varepsilon$. One can prove that at least $|\mathcal{S}| \log_2 \left( \frac{1}{\varepsilon} \right)$ bits of memory must be used to answer the approximate membership query if the elements in the set or queries are selected with equal probability from the universal set [20].

A Bloom filter [1] is one of the most basic data structures for approximate membership queries. It compresses the set $\mathcal{S}$ into a bit string using hash functions. This bit string and the hash functions are then used to answer the queries. To achieve an FPR of $\varepsilon$, a Bloom filter requires $|\mathcal{S}| \log_2 \left( \frac{1}{\varepsilon} \right) \log_2 e$ bits of memory. This is $\log_2 e$ times the theoretical lower bound.

Various derivatives have been proposed that achieve better memory efficiency than the original Bloom filter. The cuckoo filter [21] is more memory efficient than the original Bloom filter and supports dynamic addition and removal of elements. Pagh et al. [22] proposed a replacement for Bloom filter that achieves a theoretical lower bound. Various other memory-efficient filters exist, including the vacuum filter [23], xor filter [24], and ribbon filter [25]. However, these derivatives do not consider the structure of the distribution and thus cannot take advantage of it.

Kraska et al.[10] proposed using a machine learning model as a prefilter of a backup Bloom filter. Ada-BF [11] extended this design and proposed to exploit the scores output by the machine learning model. PLBF [14] uses a design similar to that of Ada-BF but introduces fewer heuristics for optimization than Ada-BF. Mitzenmacher [12] proposed an LBF that "sandwiches" a machine learning model with two Bloom filters. This achieves better memory efficiency than the original LBF but can actually be interpreted as a special case of PLBF.

# 7   Limitation and Future Work

As explained in Section 4, it is somewhat intuitive that $A$ is a monotone matrix or close to it, so it is also intuitive that fast PLBF++ achieves accuracy close to PLBF. Experimental results on the URLs and EMBER datasets also suggest that fast PLBF++ achieves almost the same accuracy as PLBF,

even when the score distribution is not *ideal*. This experimental rule is further supported by the results of the artificial data experiments described in the appendix. However, there is no theoretical support for the accuracy of fast PLBF++. Theoretical support for how fast PLBF++ accuracy may degrade relative to PLBF is a future issue.

Besides, it is possible to consider faster methods by making stronger assumptions than fast PLBF++. Fast PLBF++ assumes the *monotonicity* of matrix $A$. We adopted this assumption because matrix $A$ is proved to be *monotone* under intuitive assumptions about the data distribution. However, by assuming stronger assumptions about matrix $A$, the computational complexity of the construction could be further reduced. For example, if matrix $A$ is *totally monotone*, the matrix problem for matrix $A$ can be solved in $\mathcal{O}(N)$ using algorithms such as [16, 26]. Using such existing DP algorithms is a promising direction toward even faster or more accurate methods, and is a future work.

## 8 Conclusion

PLBF is an outstanding LBF that can effectively utilize the distribution of the set and queries captured by a machine learning model. However, PLBF is computationally expensive to construct. We proposed fast PLBF and fast PLBF++ to solve this problem. Fast PLBF is superior to PLBF because fast PLBF constructs exactly the same data structure as PLBF but does so faster. Fast PLBF++ is even faster than fast PLBF and achieves almost the same accuracy as PLBF and fast PLBF. These proposed methods have greatly expanded the range of applications of PLBF.

## Acknowledgments and Disclosure of Funding

We thank the anonymous reviewers for their constructive comments. This work was supported by JST AIP Acceleration Research JPMJCR23U2, Japan.

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

## Appendices

Appendix A details the optimization problem designed in the original PLBF paper [14] and its solution. It includes a comprehensive analysis of the optimization problem and a detailed description of how PLBF's method leads (under certain assumptions) to a solution to the optimization problem, which are not provided in [14]. Appendix B presents details of PLBF and the fast PLBF algorithm along with pseudo-code. Appendix C gives the proof of Theorem 4.3, a theorem about the accuracy of fast PLBF++. In Appendix D, we explain the trivial modifications to the PLBF framework that we made for the experiments. Appendix E describes a detailed ablation study of the PLBFs hyperparameters, $N$ and $k$. Appendix F describes the experiments on the simplified solution method presented in the PLBF paper. Appendix G describes experiments with artificial datasets to evaluate the experimental performance of fast PLBF++ in detail.

## Appendix A   Solution of the optimization problems

In the original PLBF paper [14], the analysis with mathematical expressions was conducted only for the relaxed problem, while no analysis utilizing mathematical expressions was performed for the general problem. Consequently, it was unclear which calculations were redundant, leading to the repetition of constructing similar DP tables. Hence, this appendix provides a comprehensive analysis of the general problem. It presents the optimal solution and the corresponding value of the objective function using mathematical expressions. This analysis uncovers redundant calculations, enabling the derivation of fast PLBF.

The optimization problem designed by PLBF [14] can be written as follows:

$$
\begin{aligned}
\underset{\boldsymbol{f}, \boldsymbol{t}}{\text{minimize}} \quad & \sum_{i=1}^{k} c|\mathcal{S}|G_i \log_2\left(\frac{1}{f_i}\right) \\
\text{subject to} \quad & \sum_{i=1}^{k} H_i f_i \leq F \\
& t_0 = 0 < t_1 < t_2 < \cdots < t_k = 1 \\
& f_i \leq 1 \quad (i = 1 \ldots k).
\end{aligned} \tag{7}
$$

The objective function represents the total memory usage of the backup Bloom filters. The first constraint equation represents the condition that the expected overall FPR is below $F$. $c \geq 1$ is a constant determined by the type of backup Bloom filters. $\boldsymbol{G}$ and $\boldsymbol{H}$ are determined by $\boldsymbol{t}$. Once $\boldsymbol{t}$ is fixed, we can solve this optimization problem to find the optimal $\boldsymbol{f}$ and the minimum value of the objective function. In the original PLBF paper [14], the condition $f_i \leq 1$ $(i = 1 \ldots k)$ was relaxed; however, here we conduct the analysis without relaxation. We assume $G_i > 0, H_i > 0$ $(i = 1 \ldots k)$ and $|\mathcal{S}| > 0$.

The Lagrange function is defined using the Lagrange multipliers $\boldsymbol{\mu} \in \mathbb{R}^k$ and $\nu \in \mathbb{R}$, as follows:

$$
L(\boldsymbol{f}, \boldsymbol{\mu}, \nu) = \sum_{i=1}^{k} c|\mathcal{S}|G_i \log_2\left(\frac{1}{f_i}\right) + \sum_{i=1}^{k} \mu_i(f_i - 1) + \nu\left(\sum_{i=1}^{k} H_i f_i - F\right). \tag{8}
$$

From the KKT condition, there exist $\bar{\boldsymbol{\mu}}$ and $\bar{\nu}$ in the local optimal solution $\bar{\boldsymbol{f}}$ of this optimization problem, and the following holds:

$$
\frac{\partial L}{\partial f_i}(\bar{\boldsymbol{f}}, \bar{\boldsymbol{\mu}}, \bar{\nu}) = 0 \quad (i = 1 \ldots k) \tag{9}
$$

$$
\bar{f}_i - 1 \leq 0, \quad \bar{\mu}_i \geq 0, \quad \bar{\mu}_i(\bar{f}_i - 1) = 0 \quad (i = 1 \ldots k) \tag{10}
$$

$$
\sum_{i=1}^{k} H_i \bar{f}_i - F \leq 0, \quad \bar{\nu} \geq 0, \quad \bar{\nu}\left(\sum_{i=1}^{k} H_i \bar{f}_i - F\right) = 0. \tag{11}
$$

We define $\mathcal{I}_{f=1}$ and $\mathcal{I}_{f<1}$ as $\mathcal{I}_{f=1} = \{i \mid \bar{f}_i = 1\}$ and $\mathcal{I}_{f<1} = \{i \mid \bar{f}_i < 1\}$. $\mathcal{I}_{f=1} \cup \mathcal{I}_{f<1} = \{1 \ldots k\}$ and $\mathcal{I}_{f=1} \cap \mathcal{I}_{f<1} = \varnothing$ are satisfied. Furthermore, we assume that $\mathcal{I}_{f<1} \neq \varnothing$. This means that there is at least one region that uses a backup Bloom filter with an FPR smaller than 1.

By introducing $\mathcal{I}_{f=1}$ and $\mathcal{I}_{f<1}$, Equations (9, 10, 11) can be organized as follows:

$$c|\mathcal{S}|G_i = \bar{\mu}_i + \bar{\nu}H_i, \quad \bar{\mu}_i \geq 0, \quad \bar{f}_i = 1 \quad (i \in \mathcal{I}_{f=1}) \tag{12}$$

$$c|\mathcal{S}|G_i = \bar{f}_i\bar{\nu}H_i, \quad \bar{\mu}_i = 0, \quad \bar{f}_i < 1 \quad (i \in \mathcal{I}_{f<1}) \tag{13}$$

$$\sum_{i \in \mathcal{I}_{f=1}} H_i + \sum_{i \in \mathcal{I}_{f<1}} H_i\bar{f}_i - F \leq 0, \quad \bar{\nu} \geq 0, \quad \bar{\nu}\left(\sum_{i \in \mathcal{I}_{f=1}} H_i + \sum_{i \in \mathcal{I}_{f<1}} H_i\bar{f}_i - F\right) = 0. \tag{14}$$

Here, we get $\bar{\nu} > 0$. This is because if we assume $\bar{\nu} = 0$, Equation (13) implies that $G_i = 0$ for $i \in \mathcal{I}_{f<1}(\neq \varnothing)$, which contradicts the assumption that $G_i > 0$. From $\bar{\nu} > 0$ and Equation (13),

$$\bar{f}_i = \frac{c|\mathcal{S}|}{\bar{\nu}}\frac{G_i}{H_i} \quad (i \in \mathcal{I}_{f<1}). \tag{15}$$

From Equations (14, 15) and $\bar{\nu} > 0$,

$$\sum_{i \in \mathcal{I}_{f=1}} H_i + \sum_{i \in \mathcal{I}_{f<1}} H_i\frac{c|\mathcal{S}|}{\bar{\nu}}\frac{G_i}{H_i} - F = 0 \tag{16}$$

$$\frac{c|\mathcal{S}|}{\bar{\nu}} = \frac{F - \sum_{i \in \mathcal{I}_{f=1}} H_i}{\sum_{i \in \mathcal{I}_{f<1}} G_i}. \tag{17}$$

Substituting Equations (15, 17) into the objective function of the optimization problem (7), we obtain

$$\sum_{i=1}^{k} c|\mathcal{S}|G_i \log_2\left(\frac{1}{\bar{f}_i}\right) = \sum_{i \in \mathcal{I}_{f<1}} -c|\mathcal{S}|G_i \log_2\left(\frac{F - \sum_{j \in \mathcal{I}_{f=1}} H_j}{\sum_{j \in \mathcal{I}_{f<1}} G_j}\frac{G_i}{H_i}\right) \tag{18}$$

$$= c|\mathcal{S}|(1 - G_{f=1})\log_2\left(\frac{1 - G_{f=1}}{F - H_{f=1}}\right) - c|\mathcal{S}|\sum_{i \in \mathcal{I}_{f<1}} G_i \log_2\left(\frac{G_i}{H_i}\right). \tag{19}$$

We define $G_{f=1}$ and $H_{f=1}$ as $G_{f=1} = \sum_{i \in \mathcal{I}_{f=1}} G_i$ and $H_{f=1} = \sum_{i \in \mathcal{I}_{f=1}} H_i$, respectively.

PLBF makes the following assumption:

**Assumption A.1.** For optimal $t$ and $f$, $\mathcal{I}_{f=1} = \varnothing$ or $\mathcal{I}_{f=1} = \{k\}$.

In other words, PLBF assumes that there is at most one region for which $f = 1$, and if there is one, it is the region with the highest score. PLBF then tries all possible thresholds $t_{k-1}$ (under the division of the score space).

In the following, we discuss the optimal $f$ and $t$ under the assumption that the $j$-th to $N$-th segments are clustered as the $k$-th region ($j \in \{k \dots N\}$).

In the case of $\mathcal{I}_{f=1} = \varnothing$, because $G_{f=1} = H_{f=1} = 0$, Equation (19) becomes

$$c|\mathcal{S}| \log_2\left(\frac{1}{F}\right) - c|\mathcal{S}|G_k \log_2\left(\frac{G_k}{H_k}\right) - c|\mathcal{S}|\sum_{i=1}^{k-1} G_i \log_2\left(\frac{G_i}{H_i}\right). \tag{20}$$

Meanwhile, in the case of $\mathcal{I}_{f=1} = \{k\}$, Equation (19) becomes

$$c|\mathcal{S}|(1 - G_k) \log_2\left(\frac{1 - G_k}{F - H_k}\right) - c|\mathcal{S}|\sum_{i=1}^{k-1} G_i \log_2\left(\frac{G_i}{H_i}\right). \tag{21}$$

In both cases, the terms other than the last one are constants under the assumption that the $j$-th to $N$-th segments are clustered as the $k$-th region. Therefore, the value of the objective function is minimized when the 1st to $(j-1)$-th segments are clustered in a way that maximizes $\sum_{i=1}^{k-1} G_i \log_2\left(\frac{G_i}{H_i}\right)$. Thus, the problem of finding $t$ that minimizes memory usage can be reduced to the following problem: for each $j = k \dots N$, we find a way to cluster the 1st to $(j-1)$-th segments into $k - 1$ regions that maximizes $\sum_{i=1}^{k-1} G_i \log_2\left(\frac{G_i}{H_i}\right)$.

---
**Algorithm 1** PLBF [14]
---

**Input:**
  $g \in \mathbb{R}^N$ : probabilities that the keys are contained in each segment
  $h \in \mathbb{R}^N$ : probabilities that the non-keys are contained in each segment
  $F \in (0, 1)$ : target overall FPR
  $k \in \mathbb{N}$ : number of regions
**Output:**
  $t \in \mathbb{R}^{k+1}$ : threshold boundaries of each region
  $f \in \mathbb{R}^k$ : FPRs of each region
**Algorithm:**
  THRESMAXDIVDP$(g, h, j, k)$ :
      constructs $\text{DP}_{\text{KL}}^j$ and calculates the optimal thresholds by tracing the transitions backward
      from $\text{DP}_{\text{KL}}^j[j-1][k-1]$
  OPTIMALFPR$(g, h, t, F, k)$ :
      returns the optimal FPRs for each region under the given thresholds
  SPACEUSED$(g, h, t, f)$ :
      returns the space size used by the backup Bloom filters for the given thresholds and FPRs

  MinSpaceUsed $\leftarrow \infty$
  $t_{\text{best}} \leftarrow$ None
  $f_{\text{best}} \leftarrow$ None
  **for** $j = k \ldots N$
      $t \leftarrow$ THRESMAXDIVDP$(g, h, j, k)$
      $f \leftarrow$ OPTIMALFPR$(g, h, t, F, k)$
      **if** MinSpaceUsed $>$ SPACEUSED$(g, h, t, f)$ **then**
          MinSpaceUsed $\leftarrow$ SPACEUSED$(g, h, t, f)$
          $t_{\text{best}} \leftarrow t$
          $f_{\text{best}} \leftarrow f$
  **return** $t_{\text{best}}, f_{\text{best}}$

---

## Appendix B    Algorithm details

In this appendix, we explain the algorithms for PLBF and the proposed method using pseudo-code in detail. This will help to clarify how each method differs from the others.

First, we show the PLBF algorithm in Algorithm 1. For details of OPTIMALFPR and SPACEUSED, please refer to Appendix B and Equation (2) in the original PLBF paper [14], respectively. The worst case time complexities of OPTIMALFPR and SPACEUSED are $\mathcal{O}(k^2)$ and $\mathcal{O}(k)$, respectively. As the DP table is constructed with a time complexity of $\mathcal{O}(j^2k)$ for each $j = k \ldots N$, the overall complexity is $\mathcal{O}(N^3k)$.

Next, we show the fast PLBF algorithm in Algorithm 2. The time complexity of building $\text{DP}_{\text{KL}}^N$ is $\mathcal{O}(N^2k)$, and the worst-case complexity of subsequent computations is $\mathcal{O}(Nk^2)$. Because $N > k$, the total complexity is $\mathcal{O}(N^2k)$, which is faster than $\mathcal{O}(N^3k)$ for PLBF, although fast PLBF constructs the same data structure as PLBF.

Finally, we describe the fast PLBF++ algorithm. The fast PLBF++ algorithm is nearly identical to fast PLBF, but it calculates approximated $\text{DP}_{\text{KL}}^N$ with a computational complexity of $\mathcal{O}(Nk \log N)$. Since the remaining calculations are unchanged from fast PLBF, the overall computational complexity is $\mathcal{O}(Nk \log N + Nk^2)$.

## Appendix C    Proof of Theorem 4.3

In this appendix, we present a proof of Theorem 4.3. It demonstrates that fast PLBF++ can attain the same accuracy as PLBF and fast PLBF under certain conditions.

The following lemma holds.

**Algorithm 2** Fast PLBF

   **Algorithm:**
     $\mathrm{MAXDIVDP}(\boldsymbol{g}, \boldsymbol{h}, N, k)$ :
        constructs $\mathrm{DP}_{\mathrm{KL}}^N$
     $\mathrm{THRESMAXDIV}(\mathrm{DP}_{\mathrm{KL}}^N, j, k)$ :
        calculates the optimal thresholds by tracing the transitions backward from $\mathrm{DP}_{\mathrm{KL}}^N[j-1][k-1]$

   $\mathrm{MinSpaceUsed} \leftarrow \infty$
   $\boldsymbol{t}_{\mathrm{best}} \leftarrow \mathrm{None}$
   $\boldsymbol{f}_{\mathrm{best}} \leftarrow \mathrm{None}$
   $\mathrm{DP}_{\mathrm{KL}}^N \leftarrow \mathrm{MAXDIVDP}(\boldsymbol{g}, \boldsymbol{h}, N, k)$
   **for** $j = k \ldots N$
     $\boldsymbol{t} \leftarrow \mathrm{THRESMAXDIV}(\mathrm{DP}_{\mathrm{KL}}^N, j, k)$
     $\boldsymbol{f} \leftarrow \mathrm{OPTIMALFPR}(\boldsymbol{g}, \boldsymbol{h}, \boldsymbol{t}, F, k)$
     **if** $\mathrm{MinSpaceUsed} > \mathrm{SPACEUSED}(\boldsymbol{g}, \boldsymbol{h}, \boldsymbol{t}, \boldsymbol{f})$ **then**
        $\mathrm{MinSpaceUsed} \leftarrow \mathrm{SPACEUSED}(\boldsymbol{g}, \boldsymbol{h}, \boldsymbol{t}, \boldsymbol{f})$
        $\boldsymbol{t}_{\mathrm{best}} \leftarrow \boldsymbol{t}$
        $\boldsymbol{f}_{\mathrm{best}} \leftarrow \boldsymbol{f}$
   **return** $\boldsymbol{t}_{\mathrm{best}}, \boldsymbol{f}_{\mathrm{best}}$

---

**Lemma C.1.** *Let $u_1$, $u_2$, $v_1$, and $v_2$ be real numbers satisfying*

$$0 < u_1 < u_2 \tag{22}$$

$$0 < v_1 < v_2 \tag{23}$$

$$\frac{u_1}{v_1} \geq \frac{u_2}{v_2}. \tag{24}$$

*Furthermore, we define the function $D(x, y)$ for real numbers $x > 0$ and $y > 0$ as follows:*

$$D(x, y) = \left\{ (u_1 + x) \log_2 \left( \frac{u_1 + x}{v_1 + y} \right) - u_1 \log_2 \left( \frac{u_1}{v_1} \right) \right\} - \left\{ (u_2 + x) \log_2 \left( \frac{u_2 + x}{v_2 + y} \right) - u_2 \log_2 \left( \frac{u_2}{v_2} \right) \right\}. \tag{25}$$

*If $\frac{x}{y} \geq \frac{u_1}{v_1}$ holds, then $D(x, y) \geq 0$.*

*Proof.*

$$\frac{\partial D}{\partial x} = \log_2 \left( \frac{u_1 + x}{v_1 + y} \right) - \log_2 \left( \frac{u_2 + x}{v_2 + y} \right) \tag{26}$$

$$\frac{\partial D}{\partial y} = -\frac{u_1 + x}{v_1 + y} + \frac{u_2 + x}{v_2 + y}. \tag{27}$$

From Equations (22, 23, 24), when $0 < x$, $0 < y$, and $\frac{x}{y} \geq \frac{u_1}{v_1}$, we obtain

$$\frac{u_1 + x}{v_1 + y} \geq \frac{u_2 + x}{v_2 + y}. \tag{28}$$

Therefore,

$$\frac{\partial D}{\partial x} \geq 0 \tag{29}$$

$$\frac{\partial D}{\partial y} \leq 0. \tag{30}$$

Thus,

$$\inf_{0<x,0<y,\frac{x}{y}\geq\frac{u_1}{v_1}} D(x,y) \geq \inf_{0<x,0<y,\frac{x}{y}=\frac{u_1}{v_1}} D(x,y) \tag{31}$$

$$= \inf_{z>0} D(zu_1, zv_1) \tag{32}$$

$$= \inf_{z>0} \left\{ zu_1 \log_2\left(\frac{zu_1}{zv_1}\right) + u_2 \log_2\left(\frac{u_2}{v_2}\right) - (zu_1 + u_2)\log_2\left(\frac{zu_1 + u_1}{zv_1 + v_1}\right) \right\} \tag{33}$$

$$\geq 0. \tag{34}$$

The transformation to Equation (34) is due to Jensen's inequality [27]. This shows that if $\frac{x}{y} \geq \frac{u_1}{v_1}$, then $D(x,y) \geq 0$. □

*Proof.* We prove Theorem 4.3 by contradiction.

Assume that the $(N-1) \times (N-1)$ matrix $A$ is not a *monotone matrix*. That is, we assume that there exists $p$ $(1 \leq p \leq N-2)$ such that $J(p) > J(p+1)$. Let $J(i)$ be the smallest $j$ such that $A_{ij}$ equals the maximum of the $i$-th row of $A$. We define $a := J(p)$ and $a' := J(p+1)$ $(a > a'$ from the assumption for contradiction).

From the definitions of $A$ and $J(i)$, we obtain the following:

$$\mathrm{DP_{KL}}[p][q] = \mathrm{DP_{KL}}[a-1][q-1] + d_{\mathrm{KL}}(a,p) \tag{35}$$

$$\mathrm{DP_{KL}}[p+1][q] = \mathrm{DP_{KL}}[a'-1][q-1] + d_{\mathrm{KL}}(a',p+1) \tag{36}$$

$$\mathrm{DP_{KL}}[p][q] > \mathrm{DP_{KL}}[a'-1][q-1] + d_{\mathrm{KL}}(a',p) \tag{37}$$

$$\mathrm{DP_{KL}}[p+1][q] \geq \mathrm{DP_{KL}}[a-1][q-1] + d_{\mathrm{KL}}(a,p+1). \tag{38}$$

Using Equations (35, 36, 38), we evaluate the right side of Equation (37) minus the left side of Equation (37).

$$\mathrm{DP_{KL}}[a'-1][q-1] + d_{\mathrm{KL}}(a',p) - \mathrm{DP_{KL}}[p][q] \tag{39}$$

$$= \{\mathrm{DP_{KL}}[p+1][q] - d_{\mathrm{KL}}(a',p+1)\} + d_{\mathrm{KL}}(a',p) - \{\mathrm{DP_{KL}}[a-1][q-1] + d_{\mathrm{KL}}(a,p)\} \tag{40}$$

$$= \{\mathrm{DP_{KL}}[p+1][q] - \mathrm{DP_{KL}}[a-1][q-1]\} + d_{\mathrm{KL}}(a',p) - d_{\mathrm{KL}}(a',p+1) - d_{\mathrm{KL}}(a,p) \tag{41}$$

$$\geq d_{\mathrm{KL}}(a,p+1) + d_{\mathrm{KL}}(a',p) - d_{\mathrm{KL}}(a',p+1) - d_{\mathrm{KL}}(a,p) \tag{42}$$

$$= \{d_{\mathrm{KL}}(a,p+1) - d_{\mathrm{KL}}(a,p)\} - \{d_{\mathrm{KL}}(a',p+1) - d_{\mathrm{KL}}(a',p)\}. \tag{43}$$

Equations (35, 36) are used for the transformation to Equation (40), and Equation (42) is used for the transformation to Equation (38).

We define $u_1$, $u_2$, $v_1$, $v_2$, $x$, and $y$ as follows:

$$u_1 := \sum_{i=a}^{p} g_i, \quad u_2 := \sum_{i=a'}^{p} g_i, \quad x := g_{p+1},$$

$$v_1 := \sum_{i=a}^{p} h_i, \quad v_2 := \sum_{i=a'}^{p} h_i, \quad y := h_{p+1}. \tag{44}$$

From the conditions of *ideal score distribution* and $a' < a$, the followings hold:

$$0 < u_1 < u_2 \tag{45}$$

$$0 < v_1 < v_2 \tag{46}$$

$$\frac{x}{y} \geq \frac{u_1}{v_1} \geq \frac{u_2}{v_2} \tag{47}$$

$$0 < x, y. \tag{48}$$

Therefore, from Lemma C.1,

$$\left\{ (u_1 + x)\log_2\left(\frac{u_1+x}{v_1+y}\right) - u_1 \log_2\left(\frac{u_1}{v_1}\right) \right\} - \left\{ (u_2 + x)\log_2\left(\frac{u_2+x}{v_2+y}\right) - u_2 \log_2\left(\frac{u_2}{v_2}\right) \right\} \tag{49}$$

$$= \{d_{\mathrm{KL}}(a,p+1) - d_{\mathrm{KL}}(a,p)\} - \{d_{\mathrm{KL}}(a',p+1) - d_{\mathrm{KL}}(a',p)\} \tag{50}$$

$$\geq 0. \tag{51}$$

---

**Algorithm 3** OptimalFPR_F (Algorithm 1 in [14])

---

**Input:**
    $\boldsymbol{G} \in \mathbb{R}^N$ : probabilities that the keys are contained in each region
    $\boldsymbol{H} \in \mathbb{R}^N$ : probabilities that the non-keys are contained in each region
    $F \in (0,1)$ : target overall FPR
    $k \in \mathbb{N}$ : number of regions
**Output:**
    $\boldsymbol{f} \in \mathbb{R}^k$ : FPRs of each region

  **for** $i = 1 \ldots k$
    $f_i \leftarrow \frac{G_i \cdot F}{H_i}$
  **while** $\exists\, i : f_i > 1$
    **for** $i = 1 \ldots k$
      **if** $f_i > 1$ **then**
        $f_i \leftarrow 1$
    $G_{\text{sum}} \leftarrow 0$
    $H_{\text{sum}} \leftarrow 0$
    **for** $i = 1 \ldots k$
      **if** $f_i = 1$ **then**
        $G_{\text{sum}} \leftarrow G_{\text{sum}} + G_i$
        $H_{\text{sum}} \leftarrow H_{\text{sum}} + H_i$
    **for** $i = 1 \ldots k$
      **if** $f_i < 1$ **then**
        $f_i \leftarrow \frac{G_i \cdot (F - H_{\text{sum}})}{H_i \cdot (1 - G_{\text{sum}})}$
  **return** $\boldsymbol{f}$

---

Then, from Equations (39 ... 43), we obtain

$$\text{DP}_{\text{KL}}[a'-1][q-1] + d_{\text{KL}}(a',p) - \text{DP}_{\text{KL}}[p][q] \geq 0. \tag{52}$$

However, this contradicts Equation (37). Hence, Theorem 4.3 is proved.   □

## Appendix D   Modification of the PLBF framework

In this appendix, we describe the framework modifications we made to PLBF in our experiments. In the original PLBF paper [14], the optimization problem was designed to minimize the amount of memory usage under a given target false positive rate (Equation (7)). However, this framework makes it difficult to compare the results of different methods and hyperparameters. Therefore, we designed the following optimization problem, which minimizes the expected false positive rate under a given memory usage condition:

$$
\begin{aligned}
\underset{\boldsymbol{f},\boldsymbol{t}}{\text{minimize}} \quad & \sum_{i=1}^{k} H_i f_i \\
\text{subject to} \quad & \sum_{i=1}^{k} c|\mathcal{S}| G_i \log_2\left(\frac{1}{f_i}\right) \leq M \\
& t_0 = 0 < t_1 < t_2 < \cdots < t_k = 1 \\
& f_i \leq 1 \quad (i = 1 \ldots k),
\end{aligned}
\tag{53}
$$

where $M$ is a parameter that is set by the user to determine the upper bound of memory usage and is set by the user.

Analyzing as in Appendix A, we find that in order to find the optimal thresholds $\boldsymbol{t}$, we need to find a way to cluster the 1st to $(j-1)$-th segments into $k-1$ regions that maximizes $\sum_{i=1}^{k-1} G_i \log_2\left(\frac{G_i}{H_i}\right)$ for each $j = k \ldots N$. We can compute this using the same DP algorithm as in the original framework.

---

**Algorithm 4** OptimalFPR_M

---

**Input:**
$\quad \boldsymbol{G} \in \mathbb{R}^N$ : probabilities that the keys are contained in each region
$\quad \boldsymbol{H} \in \mathbb{R}^N$ : probabilities that the non-keys are contained in each region
$\quad M \in \mathbb{R}$ : upper bound of the total memory usage of backup Bloom filters
$\quad k \in \mathbb{N}$ : number of regions
**Output:**
$\quad \boldsymbol{f} \in \mathbb{R}^k$ : FPRs of each region

$K_{\text{sum}} \leftarrow 0$
**for** $i = 1 \ldots k$
$\quad K_{\text{sum}} \leftarrow K_{\text{sum}} + G_i \log_2 \left( \frac{G_i}{H_i} \right)$
$\beta \leftarrow \frac{M + c|\mathcal{S}|K_{\text{sum}}}{c|\mathcal{S}|}$
**for** $i = 1 \ldots k$
$\quad f_i \leftarrow 2^{-\beta} \frac{G_i}{H_i}$
**while** $\exists \, i : f_i > 1$
$\quad$ **for** $i = 1 \ldots k$
$\quad\quad$ **if** $f_i > 1$ **then**
$\quad\quad\quad f_i \leftarrow 1$
$\quad G_{\text{sum}} \leftarrow 0$
$\quad H_{\text{sum}} \leftarrow 0$
$\quad K_{\text{sum}} \leftarrow 0$
$\quad$ **for** $i = 1 \ldots k$
$\quad\quad$ **if** $f_i = 1$ **then**
$\quad\quad\quad G_{\text{sum}} \leftarrow G_{\text{sum}} + G_i$
$\quad\quad\quad H_{\text{sum}} \leftarrow H_{\text{sum}} + H_i$
$\quad\quad$ **else**
$\quad\quad\quad K_{\text{sum}} \leftarrow K_{\text{sum}} + G_i \log_2 \left( \frac{G_i}{H_i} \right)$
$\quad \beta \leftarrow \frac{M + c|\mathcal{S}|K_{\text{sum}}}{c|\mathcal{S}|(1 - G_{\text{sum}})}$
$\quad$ **for** $i = 1 \ldots k$
$\quad\quad$ **if** $f_i < 1$ **then**
$\quad\quad\quad f_i \leftarrow 2^{-\beta} \frac{G_i}{H_i}$
**return** $\boldsymbol{f}$

---

Also similar to Appendix A, introducing $\mathcal{I}_{f=1}$ and $\mathcal{I}_{f<1}$ for analysis, it follows that the optimal false positive rates $\bar{\boldsymbol{f}}$ is

$$\bar{f}_i = 2^{-\beta} \frac{G_i}{H_i} \quad (i \in \mathcal{I}_{f<1}), \tag{54}$$

where

$$\beta = \frac{M + c|\mathcal{S}| \sum_{i \in \mathcal{I}_{f<1}} G_i \log_2 \left( \frac{G_i}{H_i} \right)}{c|\mathcal{S}| \left( 1 - G_{f=1} \right)}. \tag{55}$$

In the original framework, the sets $\mathcal{I}_{f=1}$ and $\mathcal{I}_{f<1}$ are obtained by repeatedly solving the relaxed problem and setting $f_i$ to 1 for regions where $f_i > 1$ (for details of this algorithm, please refer to the Section 3.3.3 in the original PLBF paper [14]). In our framework, we can find the sets $\mathcal{I}_{f=1}$ and $\mathcal{I}_{f<1}$ in the same way.

The pseudo-code for OPTIMALFPR_F in the original PLBF paper [14] and OPTIMALFPR_M, which is conditioned on the memory usage $M$, is shown in Algorithm 3 and Algorithm 4. For both functions, the best-case complexity is $\mathcal{O}(k)$, and the worst-case complexity is $\mathcal{O}(k^2)$.

Also, the fast PLBF algorithm for the case conditioned by memory usage is shown in Algorithm 5. It is basically the same as the original PLBF Algorithm 2, but instead of using the SPACEUSED function, the EXPECTEDFPR function is used here. Then, the algorithm selects the case when the expected overall false positive rate calculated using the training data is minimized.

**Algorithm 5** Fast PLBF conditioned by memory usage

---

**Algorithm:**
   $\text{MaxDivDP}(\boldsymbol{g}, \boldsymbol{h}, N, k)$ :
      constructs $\text{DP}_{\text{KL}}^N$
   $\text{ThresMaxDiv}(\text{DP}_{\text{KL}}^N, j, k)$ :
      calculates the optimal thresholds by tracing the transitions backward from $\text{DP}_{\text{KL}}^N[j-1][k-1]$
   $\text{OptimalFPR\_M}(\boldsymbol{g}, \boldsymbol{h}, \boldsymbol{t}, M, k)$ :
      returns the optimal FPRs for each region under the given thresholds and the memory usage
   $\text{ExpectedFPR}(\boldsymbol{g}, \boldsymbol{h}, \boldsymbol{t}, \boldsymbol{f})$ :
      returns the expected overall false positive rate for the given thresholds and FPRs

   $\text{MinExpectedFPR} \leftarrow \infty$
   $\boldsymbol{t}_{\text{best}} \leftarrow \text{None}$
   $\boldsymbol{f}_{\text{best}} \leftarrow \text{None}$
   $\text{DP}_{\text{KL}}^N \leftarrow \text{MaxDivDP}(\boldsymbol{g}, \boldsymbol{h}, N, k)$
   **for** $j = k \ldots N$
      $\boldsymbol{t} \leftarrow \text{ThresMaxDiv}(\text{DP}_{\text{KL}}^N, j, k)$
      $\boldsymbol{f} \leftarrow \text{OptimalFPR\_M}(\boldsymbol{g}, \boldsymbol{h}, \boldsymbol{t}, M, k)$
      **if** $\text{MinExpectedFPR} > \text{ExpectedFPR}(\boldsymbol{g}, \boldsymbol{h}, \boldsymbol{t}, \boldsymbol{f})$ **then**
         $\text{MinExpectedFPR} \leftarrow \text{ExpectedFPR}(\boldsymbol{g}, \boldsymbol{h}, \boldsymbol{t}, \boldsymbol{f})$
         $\boldsymbol{t}_{\text{best}} \leftarrow \boldsymbol{t}$
         $\boldsymbol{f}_{\text{best}} \leftarrow \boldsymbol{f}$
   **return** $\boldsymbol{t}_{\text{best}}, \boldsymbol{f}_{\text{best}}$

---

## Appendix E    Additional ablation study for hyperparameters

This appendix details the ablation studies for the hyperparameters $N$ and $k$. In this appendix, we discuss the results of the ablation studies that were not presented in Section 5.3. The results confirm that the expected false positive rate decreases monotonically as the hyperparameters $N$ and $k$ increase. We also confirm that the proposed methods are superior to PLBF in terms of hyperparameter determination.

First, we confirm that the expected false positive rate decreases monotonically as $N$ or $k$ increases. The expected false positive rate is computed using training data (objective function in Equation (53)). Figure 14 shows the expected false positive rate at various $N$ and $k$. The ablation study for $N$ is done with $k$ fixed to 5. The ablation study for $k$ is done with $N$ fixed to 1,000.

The expected false positive rate decreases monotonically as $N$ or $k$ increases for all datasets and methods. However, as observed in Section 5.3, the false positive rate at test time does not necessarily decrease monotonically as $N$ or $k$ increases. Also, the construction time increases as $N$ or $k$ increases. Therefore, it is necessary to set appropriate $N$ and $k$ when practically using PLBFs.

Therefore, we next performed a comprehensive experiment at various $N$ and $k$. Figure 15 shows the construction time and false positive rate for each method when the memory usage of the backup bloom filter is fixed at 500 Kb, $N$ is set to $8, 16, \ldots, 1024$, and $k$ is set to $3, 5, 10$, and $20$. The curves connect the same $k$ and different $N$ results.

The results show that the proposed methods are superior to PLBF because it is easier to set hyperparameters to construct accurate data structures in a shorter time. PLBF can achieve the same level of accuracy with the same construction time as fast PLBF and fast PLBF++ if the hyperparameters $N$ and $k$ are set well. However, it is not apparent how $N$ and $k$ should be set when applying the algorithm to unseen data. If $N$ or $k$ is set too large, the construction time will be very long, and if $N$ or $k$ is set too small, the false positive rate will be large. The proposed methods can construct the data structure quickly even if the hyperparameters $N$ and $k$ are too large. Therefore, by using the proposed methods instead of PLBF and setting $N$ and $k$ relatively large, it is possible to quickly and reliably construct a data structure with reasonable accuracy.

Furthermore, we performed an ablation study using fast PLBF to obtain guidelines for setting the hyperparameters $N$ and $k$ (Figure 16). We observed false positive rates for $N =$

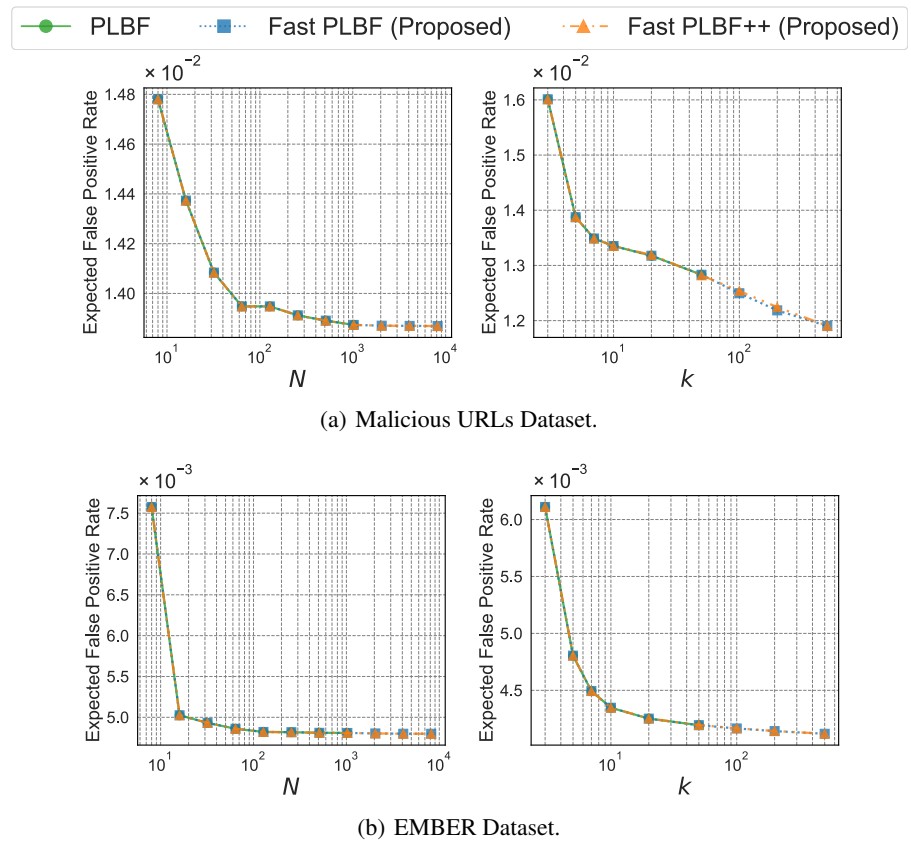

(a) Malicious URLs Dataset.

(b) EMBER Dataset.

Figure 14: Expected false positive rate computed with training data at various $N$ or $k$. The memory usage of the backup bloom filter is 500 Kb. The ablation studies for $N$ are done with $k$ fixed to 5 (left figures). The ablation studies for $k$ are done with $N$ fixed to 1,000 (right figures).

$20, 50, 100, 200, 300, \ldots, 1500$ and $k = 50, 100, 200, 300, \ldots, 1400$ with $k < N$. For the Malicious URLs Dataset, the false positive rate was lowest when $N = 1300$ and $k = 700$ (in this case, the fast PLBF construction is 333 times faster than PLBF). And for the Ember dataset, the false positive rate was lowest when $N = 500$ and $k = 50$ (in this case, the fast PLBF construction is 45 times faster than PLBF). The Malicious URLs Dataset showed an almost monotonous decrease in the false positive rate as $N$ and $k$ increased, while the Ember Dataset showed almost no such trend. This is thought to be due to a kind of overlearning that occurs as $N$ and $k$ are increased in the Ember Dataset. Although a method for determining the appropriate $N$ and $k$ has not yet been established, we suggest setting $N \sim 1000$ and $k \sim 100$. This is because the false positive rate is considered to be small enough at this level, and the proposed method can be constructed in a few tens of seconds. In any case, the proposed method is useful because it is faster than the original PLBF for any hyperparameters $N$ and $k$.

## Appendix F    Experiments on PLBF solution to relaxed problem

The PLBF paper [14] proposes two methods: a simple method and a complete method. In this appendix, we conducted experiments on the simple method and evaluated the difference in accuracy compared to the complete method. The results revealed that, in certain cases, this method exhibited significantly lower accuracy than the complete method. It was then confirmed that fast PLBF and fast PLBF++ are superior to the simple method in terms of the trade-off between construction time and accuracy.

The simpler method, described in section 3.3.3 of [14], does not consider the condition that $f_i \leq 1$ $(i = 1 \ldots k)$ when finding the optimal $t$. In other words, this method solves the relaxed problem

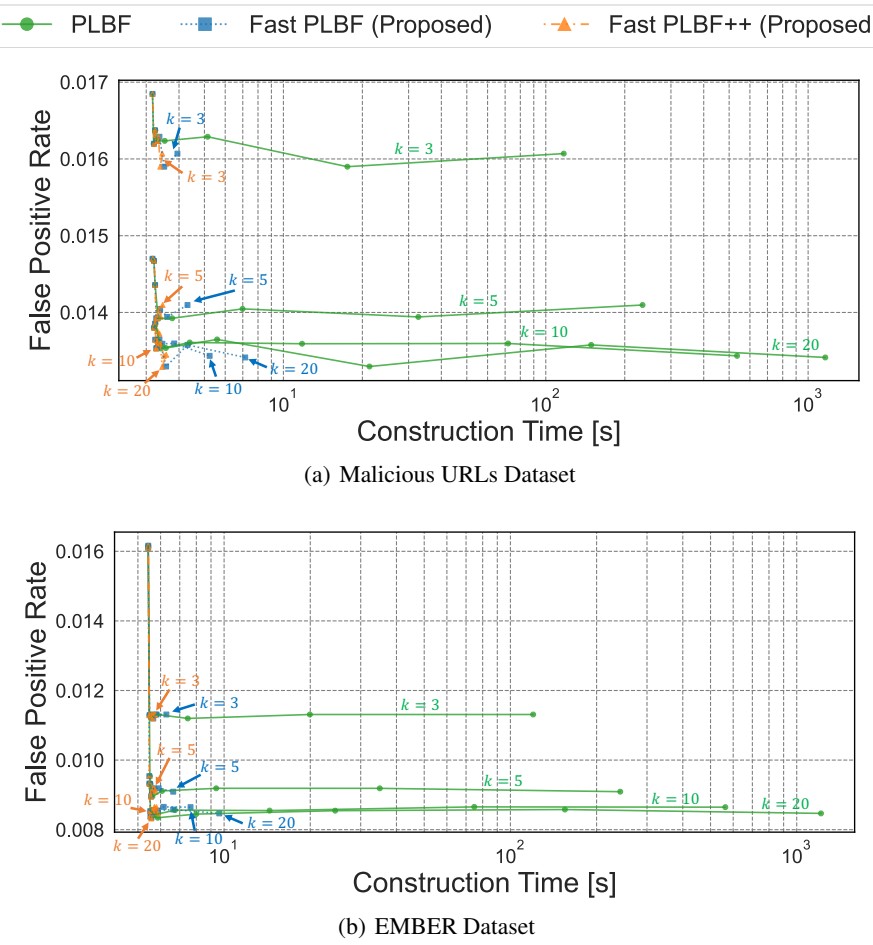

(a) Malicious URLs Dataset

(b) EMBER Dataset

Figure 15: Construction time and FPR for various $N$ and $k$. The curves connect the same $k$ and different $N$ results. The memory usage of the backup bloom filter is 500 Kb.

that does not consider the condition $f_i \leq 1$ $(i = 1 \ldots k)$. The complete method uses the first method repeatedly as a subroutine to solve the general problem and find the optimal $t$. Here, the time complexity of the method for the relaxed problem is $\mathcal{O}(N^2 k)$, and that of the method for the general problem is $\mathcal{O}(N^3 k)$.

We experimented with the simpler method. Figure 17 shows the construction time (the hyperparameters for PLBFs were set to $N = 1,000$ and $k = 5$). The PLBF for the relaxed problem has about the same construction time as our proposed fast PLBF. This is because the time complexity for constructing the two methods is $\mathcal{O}(N^2 k)$ for both. Figure 18 shows the trade-off between memory usage and FPR. On the URL dataset, the method for the relaxed problem is almost as accurate as the method for the general problem. At most, it has a false positive rate of only 1.06 times greater. On the other hand, on the EMBER dataset, the method for the relaxed problem is significantly less accurate than the method for the general problem, especially when memory usage is small. It has up to 1.48 times higher false positive rate than the complete method.

In summary, the method for the relaxed problem can construct the data structure faster than the method for the general problem, but the accuracy can be significantly worse, especially when memory usage is small. Our proposed fast PLBF has the same accuracy as the complete PLBF in almost the same construction time as the method for the relaxed problem. Our fast PLBF++ can construct the data structure in even less time than the method for the relaxed problem, and the accuracy degradation is relatively small.

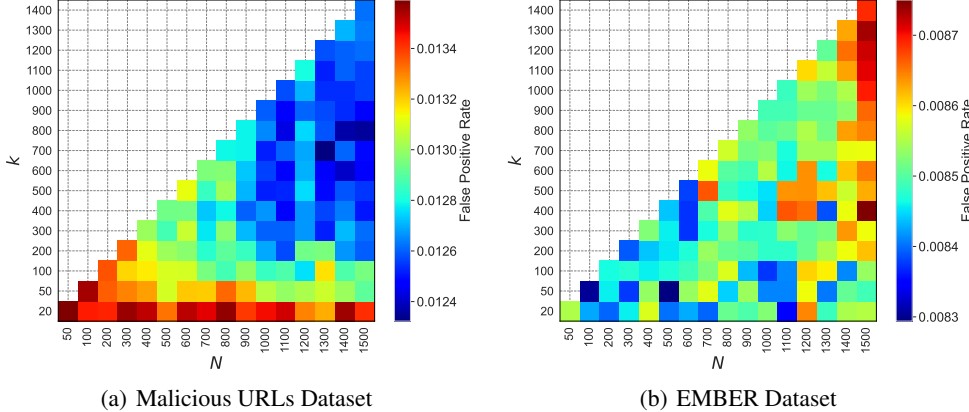

| (a) Malicious URLs Dataset | (b) EMBER Dataset |

Figure 16: FPR for $N = 50, 100, 200, 300, \ldots, 1400$ and $k = 20, 50, 100, 200, 300, \ldots, 1500$ with $k < N$.

## Appendix G   Experiments with artificial datasets

We experimented with artificial datasets. Fast PLBF++ is faster to construct than fast PLBF. However, fast PLBF++ is not theoretically guaranteed to be as accurate as PLBF, except when the score distribution is "ideally monotonic." Therefore, we evaluated the accuracy of fast PLBF++ by creating a variety of datasets, ranging from data with monotonicity to data with little monotonicity. The results show that the smaller the monotonicity of the score distribution, the larger the difference in accuracy between fast PLBF++ and PLBF. We also observed that in most cases, the false positive rate for fast PLBF++ is within 1.1 times that of PLBF, but in cases where there is little monotonicity in the score distribution, the false positive rate of PLBF++ can be up to 1.85 times that of PLBF.

Here, we explain the process of creating an artificial dataset, which consists of two steps. First, as in the original PLBF paper [14], the key and non-key score distribution is generated using the Zipfian distribution. Figure 19(a) shows a histogram of the distribution of key and non-key scores at this time, and Figure 20(a) shows $g_i/h_i$ $(i = 1 \ldots N)$ when $N = 1,000$. This score distribution is ideally monotonic. Next, we perform *swaps* to add non-monotonicity to the score distribution. *Swap* refers to changing the scores of the elements in the two adjacent segments so that the number of keys and non-keys in the two segments are swapped. Namely, an integer $i$ is randomly selected from $\{1, 2, \ldots, N-1\}$, and the scores of elements in the $i$-th segment are changed so that they are included in the $(i + 1)$-th segment, and the scores of elements in the $(i + 1)$-th segment are changed to include them in the $i$-th segment. Figures 19(b) to 19(i) and Figures 20(b) to 20(i) show the histograms of the score distribution and $g_i/h_i$ $(i = 1 \ldots N)$ for $10, 10^2, \ldots, 10^8$ swaps, respectively. It can be seen that as the number of swaps increases, the score distribution becomes more non-monotonic. For each case of the number of swaps, 10 different datasets were created using 10 different seeds.

Figure 21 shows the accuracy of each method for each number of swaps, with the seed set to 0. Hyperparameters for PLBFs are set to $N = 1,000$ and $k = 5$. It can be seen that fast PLBF and fast PLBF++ achieve better Pareto curves than the other methods for all datasets. It can also be seen that the higher the number of swaps, the more often there is a difference between the accuracy of fast PLBF++ and fast PLBF.

Figure 22 shows the difference in accuracy between fast PLBF++ and fast PLBF for each swap count. Here, the "relative false positive rate" is the false positive rate of fast PLBF++ divided by that of PLBF constructed with the same hyperparameters and conditions (note that fast PLBF constructs the same data structure as PLBF, so the false positive rate of fast PLBF is the same as that of PLBF). We created 10 different datasets for each swap count and conducted 6 experiments for each dataset and method with a memory usage of 0.25Mb, 0.5Mb, 0.75Mb, 1.0Mb, 1.25Mb, and 1.5Mb. Namely, we compared the false positive rates of fast PLBF and fast PLBF++ under 60 conditions for each swap count. The result shows that fast PLBF++ consistently achieves the same accuracy as PLBF in a total of 240 experiments where the number of swaps is $10^3$ or less. It also shows that in the experiments where the number of swaps is $10^7$ or less, the false positive rate of fast PLBF++ is less than 1.1 times

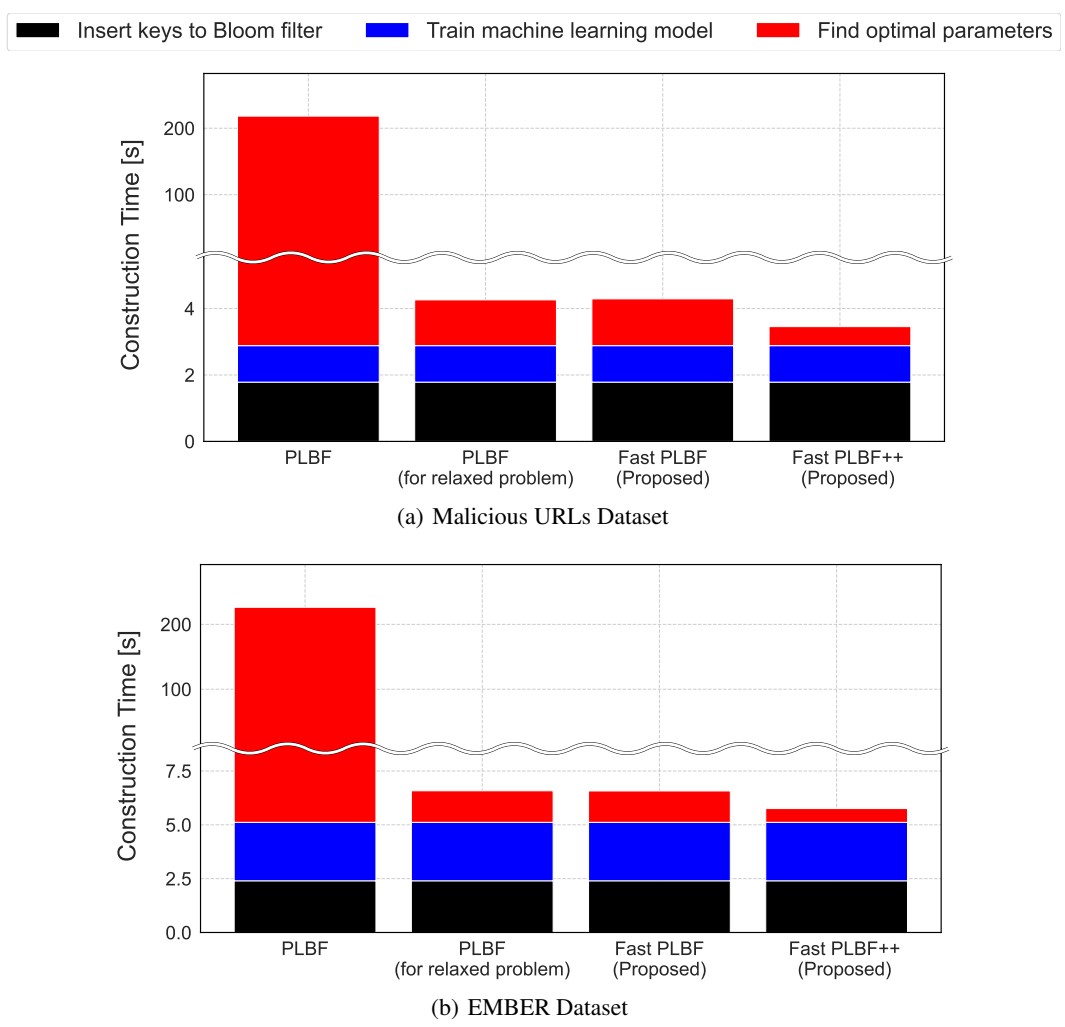

(a) Malicious URLs Dataset

(b) EMBER Dataset

Figure 17: Construction time. The memory usage of the backup bloom filter is 500 Kb, and the hyperparameters for PLBFs are $N = 1,000$ and $k = 5$.

that for PLBF, except for 14 cases. However, in the cases of $10^8$ swap counts (where there is almost no monotonicity in the score distribution), the false positive rate for fast PLBF++ is up to 1.85 times that for PLBF.

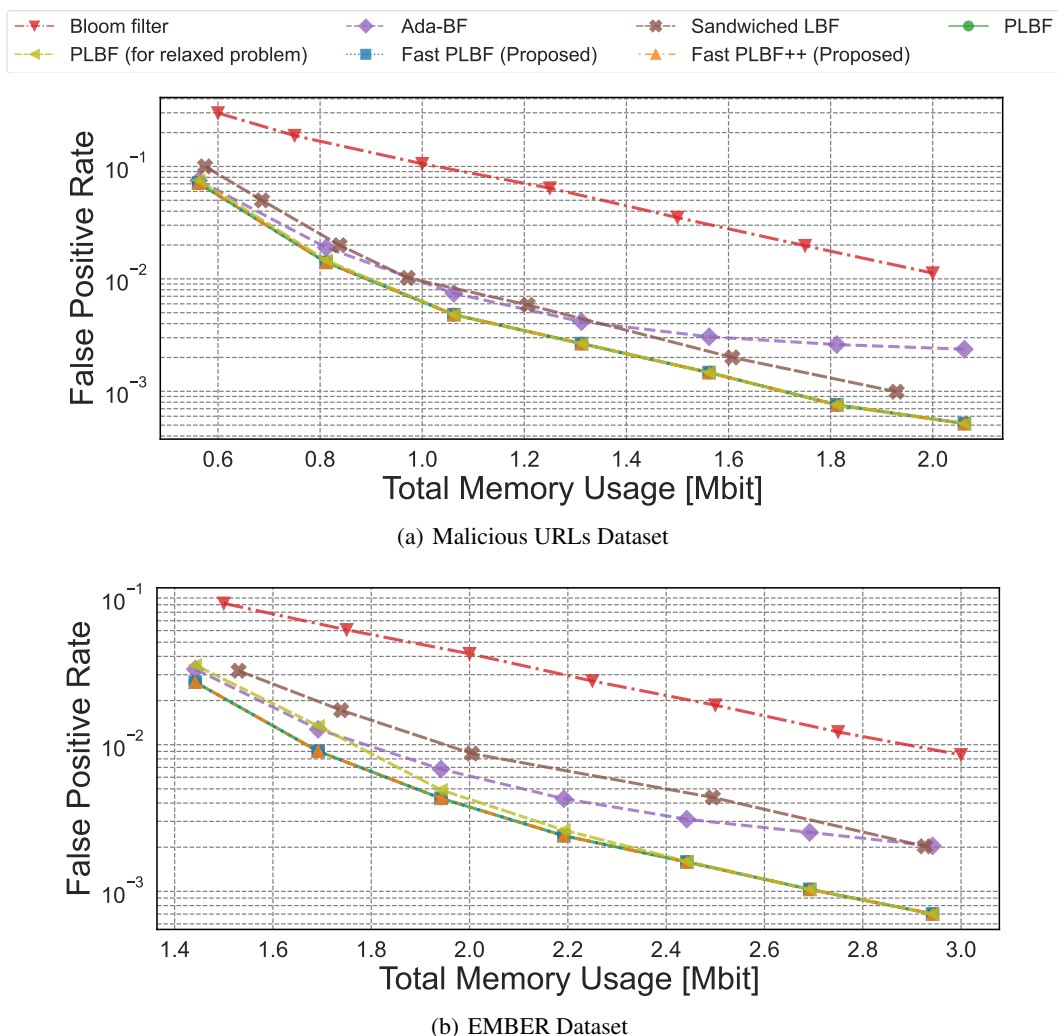

(a) Malicious URLs Dataset

(b) EMBER Dataset

Figure 18: Trade-off between memory usage and FPR. The hyperparameters for PLBFs are $N = 1,000$ and $k = 5$.

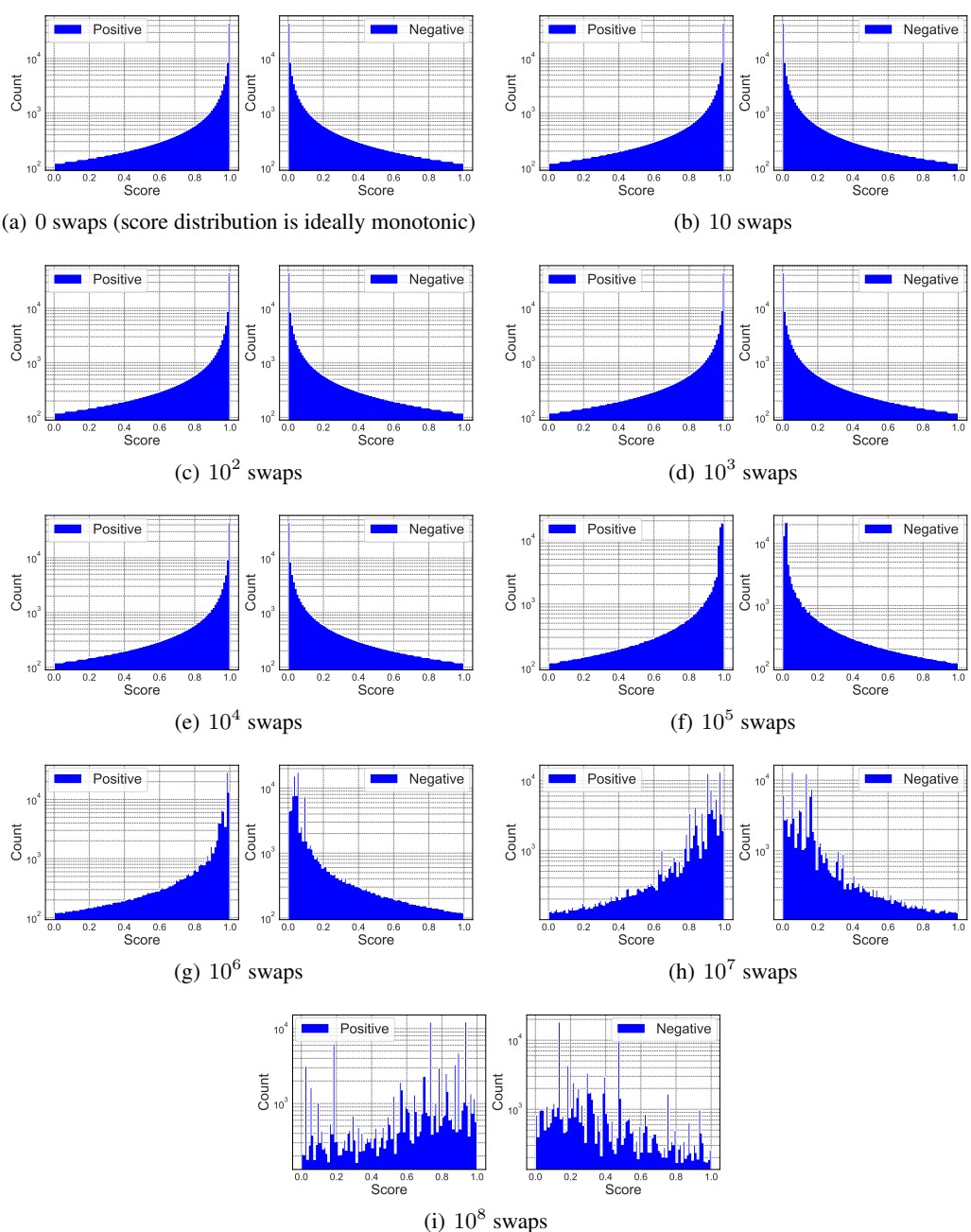

Figure 19: Score distribution histograms of artificial datasets. All figures are with the seed set to 0.

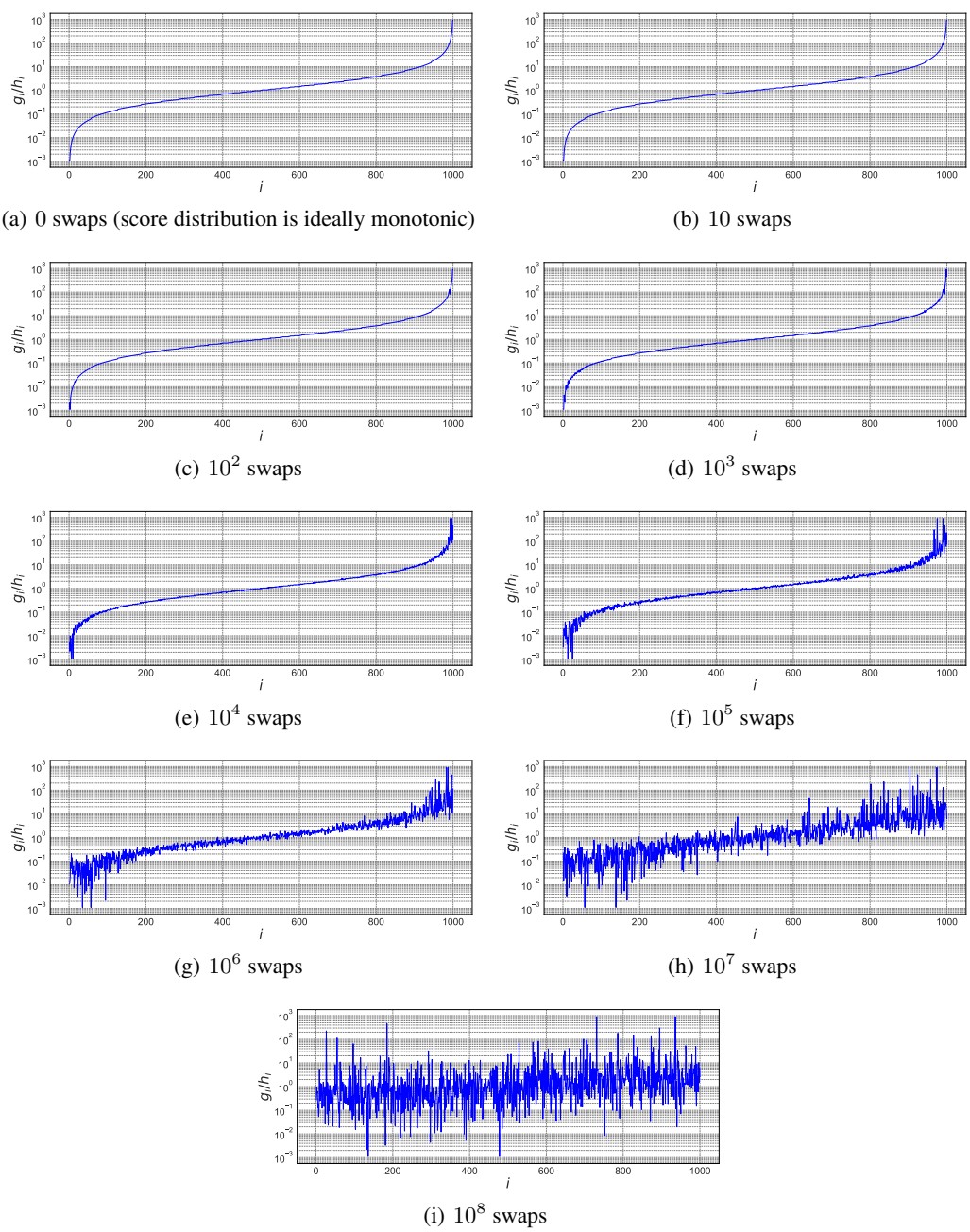

Figure 20: Ratio of key to non-key of artificial datasets. All figures are with the seed set to 0.

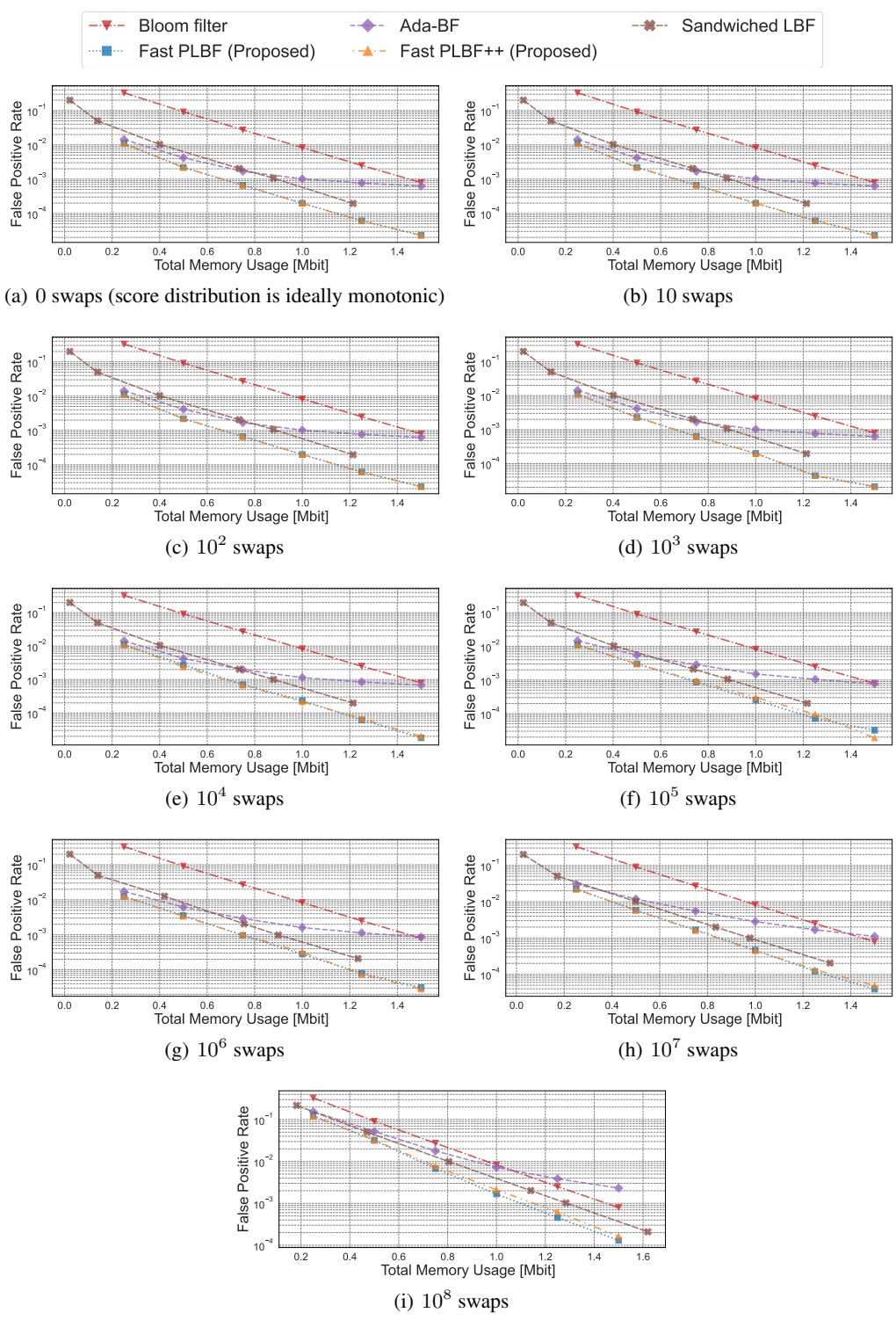

Figure 21: Trade-off between memory usage and FPR for artificial datasets. All figures are with the seed set to 0. The hyperparameters for PLBFs are $N = 1,000$ and $k = 5$.

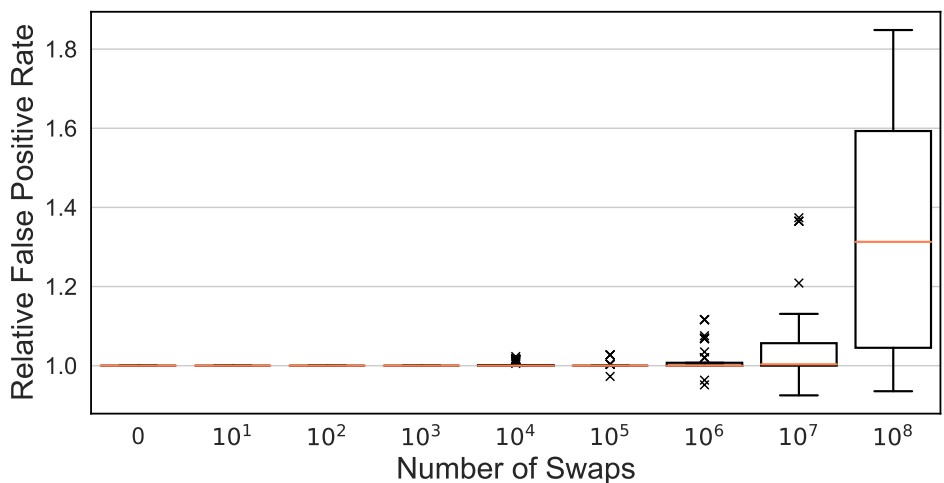

Figure 22: Distribution of the "relative false positive rate" of fast PLBF++ for each swap count. The "relative false positive rate" is the false positive rate of fast PLBF++ divided by that of PLBF constructed with the same hyperparameters and conditions.

