# OpenReview forum: "Fast Partitioned Learned Bloom Filter"
_NeurIPS.cc/2023/Conference — NeurIPS 2023 poster_

### Official Review · Reviewer_YWKv · 2023-06-27

**Soundness:** 3 good
**Presentation:** 3 good
**Contribution:** 3 good
**Rating:** 6
**Confidence:** 3

**Summary:**

The authors propose two methods to reduce the construction time of the partitioned learned Bloom filter (PLBF):
1. Fast PLBF, which can construct the same data structure as PLBF but with a smaller time complexity of O(N^2k) and 2. Fast PLBF++, which can construct the data structure with a time complexity of O(Nk log N + Nk^2) but may not necessarily construct the same data structure as PLBF. Fast PLBF++ is almost as memory efficient as PLBF.

The authors prove that fast PLBF++ has the same data structure as PLBF when the distribution satisfies a certain constraint.


**Strengths:**

1. The algorithm and presentation are straightforward and strong theoretical results are provided.

2. The author shows empirically that their construction keeps an edge in practice and can lead to faster PLBF construction.



**Weaknesses:**

The authors may provide further ablation studies on the sensitivity of hyper-parameters in practice and discuss more about societal impact.

**Questions:**

My concerns are raised above.

**Limitations:**

The authors may include a discussion on the potential negative societal impact of their work.

---

> ### Author Rebuttal · Authors · 2023-08-09
>
> We appreciate the reviewer's positive assessment of our paper. We would like to address their concerns.
>
> > The authors may provide further ablation studies on the sensitivity of hyper-parameters in practice
>
> In response to this comment, we performed an ablation study to verify the sensitivity of the hyperparameters. The results showed that changing the hyperparameters from (N,k)=(1000, 100) to some extent does not dramatically change the false positive rate. In other words, the accuracy is robust to the hyperparameters, at least around (N,k)=(1000, 100). (The reason for considering around (N,k)=(1000, 100) is that previous ablation studies have shown that setting N to about 1000 and k to about 100 is appropriate to keep the false positive rate small and the build time short. For details, see the ablation studies mentioned in response to reviewer cwdf.)
>
> Specifically, we performed a grid search for N=800, 850, 900, ..., 1200, and k=80, 85, 90, ..., 120. The memory usage of the backup Bloom filters was fixed at 500 Kb. The results are summarized as follows.
>
> In the case of URL dataset,
> - best case (minimal FPR case): when (N, k)=(950, 120), the FPR was 1.26%.
> - worst case (maximal FPR case): when (N, k)=(850, 90), the FPR was 1.31%, i.e., 1.037 times larger than the minimum.
>
> In the case of Ember dataset,
> - best case (minimal FPR case): when (N, k)=(1100, 115), the FPR was 0.830%.
> - worst case (maximal FPR case): when (N, k)=(1200, 80), the FPR was 0.870%, i.e., 1.047 times larger than the minimum.
>
> Thus, at most, the false positive rate is only 1.037 or 1.047 times larger than the best case. Therefore, around (N, k)=(1000, 100), the accuracy is robust to hyperparameters.
>
> This ablation study will be included in the final version.
>
> > The authors may … discuss more about societal impact.
>
> > The authors may include a discussion on the potential negative societal impact of their work.
>
> The applicability of fast PLBF is broad, including databases and networks. For example, as our experiments suggest, it can be used to detect blacklisted malicious URLs or files. Furthermore, it may improve cache efficiency in databases or optimize routing tables in networks, as in the case of Bloom filters. The scope of fast PLBF fully encompasses and is broader than that of PLBF. In particular, our fast PLBF is expected to have a significant advantage over the original PLBF when applied to applications where the set to be retained changes frequently. This is because in such use cases, the PLBF must be rebuilt repeatedly to maintain accuracy, and fast PLBF has a much shorter construction time than PLBF.
>
> In addition, we believe that fast PLBF is unlikely to have a negative societal impact because it speeds up the construction of PLBF while maintaining its full accuracy.
>
> In the final manuscript, we will expand on the use of the proposed methods in the practical/social applications discussed above.

---

### Official Review · Reviewer_SSjS · 2023-06-29

**Soundness:** 3 good
**Presentation:** 1 poor
**Contribution:** 2 fair
**Rating:** 5
**Confidence:** 3

**Summary:**

The authors propose faster dynamic programming variants of the learned partitioned Bloom filter data structure of Vaidya et al. that requires $O(N^3 k)$ time.
Two solutions are proposed, the first constructs the same data structure in time $O(N^2 k) $ time, and another, that constructs a potentially different structure in $O(Nk\log N+Nk^2)$ time.

**Strengths:**

A significant speedup of a recent paper on a hot and interesting topic. Also, a nice empirical gains in the construction speed.

**Weaknesses:**

In essence, the authors propose DP acceleration techniques without explaining the difference from off-the-shelf methods.
Many such solutions are known for decades (e.g., see https://courses.engr.illinois.edu/cs473/sp2016/notes/06-sparsedynprog.pdf), and while it is possible that they don't cover the exact problem, they seem very similar.

While the authors propose to use the SMAWK algorithm, it seems that the wrapper needed to get from it to the actual dynamic program might also be known.
For example, (https://www.sciencedirect.com/science/article/pii/002001909090215J) solves the following problem:

$$ E[j] = \min_{0\le i<j} D[i] + w(i,j)\ .$$

This looks nearly identical to your Equation (2), and I believe that a small massaging of the notations might yield your $O(N^2 k)$ solution.
Without explaining the difference from known DP acceleration methods, it is very hard to assess the novelty of the presented algorithms.


I am also unclear about how you set $N$ in practice.

**Questions:**

1. How do you set $N$?

2. Is the error monotone in $N$? (If so, the answer to (1) is simple -- use the largest feasible $N$. I'm not sure otherwise.)

3. Can you explain the differences between your solutions and existing DP acceleration techniques?

4. Vaidya et al. also present an algorithm with runtime $O(N^2 k)$ in their paper, under some assumptions. Can you please comment about whether this is the same conditions you use in Fast PLBF++?

5. In cases where Fast PLBF++ yields a different data structure than PLBF, can you bound the increase in error?

---

> ### Author Rebuttal · Authors · 2023-08-09
>
> We appreciate the reviewer's insightful comments. We are happy to answer the questions raised.
>
> > How do you set N?
>
> Finding the optimal N and k is difficult, but setting N to about 1000 and k to about 100 may be a good choice. The additional ablation study using two real-world datasets shows the following (for details, see the response to reviewer cwdf):
>
> - The FPR is small enough when N\~1000 and k\~100. A larger N or k seems not to cause a much further small FPR.
>
> - When N\~1000 and k\~100, the proposed methods take only a few seconds to build (although the original PLBF takes more than an hour).
>
> Although it is not sure whether N\~1000 and k\~100 make the FPR small enough for unknown datasets, our proposed methods are beneficial because they can always be constructed faster than the original PLBF for any N and k.
>
> > Is the error monotone in N? (If so, the answer to (1) is simple -- use the largest feasible N. I'm not sure otherwise.)
>
> As N increases, the accuracy improves (although not necessarily monotonically), but at the same time, construction time increases. Therefore, depending on the application, N should be determined appropriately, considering the trade-off between accuracy and construction time.
>
> In the case of the original PLBF, the construction time significantly increases as N increases, while the proposed methods do not suffer from such long construction times. Hence, it is easier to balance accuracy and construction time using the proposed methods.
>
> > Can you explain the differences between your solutions and existing DP acceleration techniques?
>
> The two methods we proposed are not novel DP acceleration methods but methods that accelerate the PLBF construction by focusing mainly on the DP part of the PLBF construction.
>
> First, fast PLBF is not a DP acceleration method; it is a method that achieves acceleration by solving a large number of “problems” collectively, whereas PLBF solves them separately. PLBF iterates over all the candidates of the rightmost region and finds the optimal thresholds (i.e., t) and false positive rates (i.e., f) for each candidate. Since there are N-k+1 candidates of the rightmost region, there are N-k+1 "problems" to solve (see Appendix C of the original PLBF paper for details). PLBF solves N-k+1 problems separately, i.e., it computes the N-k+1 DP tables separately. Fast PLBF, on the other hand, computes only one DP table and leverages it repeatedly to solve N-k+1 “problems.” We mathematically showed that the results of PLBF and fast PLBF are identical.
>
> Second, indeed fast PLBF++ introduces a faster calculation of DP and is exactly the same as fast PLBF except for the DP part. However, for this acceleration, it is necessary to discover that "monotonicity" often appears in the DP calculation for PLBF construction. We provided the following:
>
> - An intuitive explanation for the appearance of "monotonicity" (see Figure 7).
> - A theoretical proof (Theorem 4.3) that the perfect "monotonicity" appears under the assumption that the score distribution is “ideal” (this assumption is specific to the PLBF problem and is a natural assumption).
> - Experimental results indicating that fast PLBF++ achieves almost the same accuracy as (fast) PLBF in many cases, even when the score distribution is not ideal (see Section 5 and Appendix G).
>
> In conclusion, fast PLBF++ is not novel as a DP speedup method, but it is a speedup method proposed by careful analysis of the PLBF problem.
>
> We will include the above discussion in the final manuscript. In addition, we must further discuss the relationship between the proposed methods and general DP acceleration methods pointed out by the reviewer SSjS. Such a discussion will solidify the theoretical connection between our approach and existing DP acceleration methods. Thanks again for the informative and insightful comments.
>
> > Vaidya et al. also present an algorithm with runtime O(N^2k) in their paper, under some assumptions. Can you please comment about whether this is the same conditions you use in Fast PLBF++?
>
> These two assumptions are different.
>
> PLBF's $O(N^2 k)$ method assumes that "the false positive rate is less than 1 for all regions when optimal thresholds and false positive rates are set for each region" (our experiments in Appendix F show that this method can be very inaccurate compared to PLBF's $O(N^3 k)$ method).
>
> On the other hand, fast PLBF and fast PLBF++ do not make this assumption. The assumption made by fast PLBF++ is that "the fraction of positive elements among the elements in the $i$-th segment ($g_i/h_i$) increases monotonically with increasing $i$". Under this assumption, we can prove that fast PLBF++ achieves the same accuracy as PLBF. However, even when this assumption does not hold, fast PLBF++ and PLBF experimentally achieve almost the same accuracy.
>
> > In cases where Fast PLBF++ yields a different data structure than PLBF, can you bound the increase in error?
>
> As mentioned in the Limitation section of the main text, it is a future work to theoretically bound the increase in error. Currently, we have found the following:
>
> - Theoretical proof (provided in Appendix C) shows that fast PLBF++ is consistent with PLBF under ideal conditions.
> - Experiments on real datasets (in Section 5) show that fast PLBF++ consistently achieves almost the same accuracy as PLBF.
> - Experiments on many artificial data (in Appendix G) show that the accuracy of fast PLBF++ rarely seems to deteriorate catastrophically, unless the situation is very far from ideal.

---

> > ### Comment · Reviewer_SSjS · 2023-08-10
> >
> > Thank you for your rebuttal.
> >
> > Can you please relate to whether "Concave-1D" (https://www.sciencedirect.com/science/article/pii/002001909090215J) is applicable to your problem and what would the resulting runtime be?
> >
> > It solves the following DP problem:
> >
> > $$ E[j] = \min_{0\le i<j} D[i] + w(i,j)\ ,$$
> >
> > and it seems that a massaging of the notations might yield a $O(N^2 k)$ solution.
> >
> > Also, is it possible to generalize your solution to a general DP setting of the above form? What would be the requirements from $E,D,w$, and what would be the resulting runtime?
> > If you can reformulate your solution as a general DP technique it can have impact much greater than accelerating PLBF.

---

> > > ### Comment · Reviewer_SSjS · 2023-08-16
> > >
> > > Without further clarification, I'm forced to lower my score, as I don't think the paper can be accepted if we don't establish the delta from known methods.

---

> > > > ### Author Response · Authors · 2023-08-17
> > > >
> > > > We apologize for the delay in responding. We implemented concave-1d to address reviewer SSjS's concerns in detail, but it took some time. We appreciate SSjS's prompt response and are happy to address the questions raised.
> > > >
> > > > > Can you please relate to whether "Concave-1D" (https://www.sciencedirect.com/science/article/pii/002001909090215J) is applicable to your problem and what would the resulting runtime be?
> > > >
> > > > It is possible to apply "concave-1D" to the PLBF problem. By using concave-1D, the computational complexity of the DP table construction is reduced to $O(Nk)$, and the overall computational complexity becomes $O(Nk + Nk^2) = O(Nk^2)$. We implemented the method using concave-1D and measured its construction speed and accuracy. **The results showed that the speed and accuracy were about the same as Fast PLBF++.**
> > > >
> > > > The differences between the method using concave-1d and fast PLBF++ can be summarized as follows:
> > > >
> > > > - For the method using concave-1d,
> > > >   - it is guaranteed to produce the same solution as PLBF when matrix $B$ is "totally monotone".
> > > >   - its computational complexity of the DP table construction is $O(Nk)$.
> > > >   - its overall computational complexity is $O(Nk + Nk^2) = O(Nk^2)$.
> > > >
> > > > - For fast PLBF++,
> > > >   - it is guaranteed to produce the same solution as PLBF when matrix $B$ is "monotone".
> > > >   - its computational complexity of the DP table construction is $O(Nk \log N)$.
> > > >   - its overall computational complexity is $O(Nk \log N + Nk^2)$.
> > > >
> > > > In other words, from a theoretical point of view, concave-1D could be faster (i.e., less computationally expensive than fast PLBF++) but less accurate (i.e., a more stringent assumption).
> > > >
> > > > We implemented a method for constructing PLBF using concave-1D. We then evaluated it on two real-world datasets, as described in Section 5. The results showed the following:
> > > > - Construction time: The construction time is almost as long as that of Fast PLBF++ (in some cases even longer). This result is counterintuitive; concave-1D is slower than we thought. This result could be because the DP construction of Fast PLBF++ is fast enough ($O(Nk \log N)$), and the dominant factor could be the subsequent computation to obtain the optimal FPRs for each region under the given thresholds ($O(Nk^2)$). Thus, further speeding up DP construction by concave-1d ($O(Nk)$) may not provide any speedup. Also, since $\log N$ is small, the constant factor of concave-1d might be more significant.
> > > > - Accuracy: The concave-1D method is almost as accurate as Fast PLBF(++). This result is also counterintuitive; concave-1D achieves better results than we imagined.
> > > >
> > > > We would like to thank reviewer SSjS again for highlighting this point. We will include the discussed content in the final manuscript. The consideration of concave-1d provides a clear avenue for further improvement of our approach. While the short rebuttal period makes it difficult to draw definitive conclusions, we look forward to discussing this point more fully in our camera-ready manuscript
> > > >
> > > > Note that we believe the above discussion does not diminish the value of our paper. Fast PLBF++ is an important proposal because
> > > > - Fast PLBF++ makes DP computation fast enough. The jump from PLBF to Fast PLBF(++) is significant, and no one has yet achieved it.
> > > > - Fast PLBF++ assumes only "monotone", which we can intuitively explain why they often appear in this PLBF problem.

---

> > > > > ### Author Response · Authors · 2023-08-17
> > > > >
> > > > > > Also, is it possible to generalize your solution to a general DP setting of the above form? What would be the requirements from E, D, w, and what would be the resulting runtime?
> > > > >
> > > > > The divide and conquer method used in fast PLBF++ can obtain a solution for the general DP problems formulated as follows in $O(n \log n)$ time: given a real-valued function $w(i, j)$ for integers $0 \le i \le j \le n$ and an array $D[i]$ for $0 \le i \le n$, compute
> > > > >
> > > > > $$ E[j] = \min_{0\le i<j} D[i] + w(i,j) ~~~~~ \textrm{for} ~~ 1 \le j \le n.$$
> > > > >
> > > > > The result is consistent with the exact solution when $E$, $D$, and $w$ satisfy the following condition:
> > > > >
> > > > > - The matrix $B$ is "monotone", where $B$ is defined as $B[i, j] = D[i] + w(i, j)$ ($0 \le i < j \le n$). In other words, $r(j)$, the smallest row index that takes that smallest value in column $j$ of matrix $B$, is non-decreasing.
> > > > >
> > > > > When $B$ is not "monotone," this method may yield suboptimal solutions. In practical applications of fast PLBF++, $B$ is not always guaranteed to be "monotone," which could lead to suboptimal results.

---

> > > > > ### Comment · Reviewer_SSjS · 2023-08-17
> > > > >
> > > > > Dear authors,
> > > > >
> > > > > Thank you for the reply and explanations.
> > > > > I didn't expect you to implement this during the rebuttal (but rather understand the assumptions and runtime) so this is appreciated.
> > > > > It's indeed interesting that concave-1d worked reasonably well in practice.
> > > > >
> > > > > Can you please explain the difference between the "monotone" and "totally monotone" assumptions in terms of PBLF and when either would be applicable?
> > > > >
> > > > > Regarding the runtime, I guess that the values of $k,N$ and the underlying constants make the $O(Nk\log N)$ factor small compared to $O(Nk^2)$?
> > > > >
> > > > > Also, can you comment on how your solution(s) will compare with the current approaches that assume the matrix to be (not necessarily strictly-) monotone? (E.g., see Section 6.4 of https://courses.engr.illinois.edu/cs473/sp2016/notes/06-sparsedynprog.pdf; I can try to dig up the paper if you'd like to have something citable.)
> > > > >
> > > > > I'll restore my original score for now.

---

> > > > > > ### Author Response · Authors · 2023-08-18
> > > > > >
> > > > > > We thank SSjS again for the prompt response and are happy to answer the questions.
> > > > > >
> > > > > > > Can you please explain the difference between the "monotone" and "totally monotone" assumptions in terms of PBLF and when either would be applicable?
> > > > > >
> > > > > > In terms of the PLBF, "monotone" means that the optimal threshold is non-decreasing (i.e., does not move *left*) when the range of segments considered is increased by one *right* (see Figure 7 in the paper). In other words, the optimal threshold for dividing the 1 to (p+1)-th segment into q regions is the same or further to the *right* than that for dividing the 1 to p-th segment into q regions. We believe this is an intuitive enough assumption.
> > > > > >
> > > > > > Although the intuitive explanation of "totally monotone" in terms of PLBF is not yet available, we can express mathematically it as follows: for all $1 \le j < i \le N-2$,
> > > > > >
> > > > > > $$ d_\mathrm{KL}(j, i) + d_\mathrm{KL}(j+1, i+1) \ge d_\mathrm{KL}(j+1, i) + d_\mathrm{KL}(j, i+1), $$
> > > > > >
> > > > > > where $d_\mathrm{KL}$ is the function defined in equation (3) in the paper.
> > > > > >
> > > > > > As to when either would be applicable, the following holds:
> > > > > >
> > > > > > - If "totally monotone," then "monotone" (the converse does not hold). In other words, "totally monotone" is a more stringent assumption than "monotone.”
> > > > > >
> > > > > > - If the score distribution is *ideal*, then "monotone" (Theorem 4.3).
> > > > > >
> > > > > >
> > > > > > > Regarding the runtime, I guess that the values of $k,N$ and the underlying constants make the $O(Nk\log N)$ factor small compared to $O(Nk^2)$?
> > > > > >
> > > > > > We agree, and we believe it is why the method with Concave-1D is not so much faster than Fast PLBF++ experimentally.
> > > > > >
> > > > > >
> > > > > > > Also, can you comment on how your solution(s) will compare with the current approaches that assume the matrix to be (not necessarily strictly-) monotone? (E.g., see Section 6.4 of https://courses.engr.illinois.edu/cs473/sp2016/notes/06-sparsedynprog.pdf; I can try to dig up the paper if you'd like to have something citable.)
> > > > > >
> > > > > > The divide-and-conquer algorithm used by Fast PLBF++ is identical to the algorithm mentioned in the Aggarwal et al. paper [15] (and this method is presented also in Section 6.4 of https://courses.engr.illinois.edu/cs473/sp2016/notes/06-sparsedynprog.pdf). Again, Fast PLBF++ is not a novel DP acceleration method. Instead, our contribution is the significant speedup of PLBF construction by carefully observing the PLBF problem and making an intuitive assumption sufficient for speedup.

---

> > > > > > > ### Comment · Reviewer_SSjS · 2023-08-19
> > > > > > >
> > > > > > > Thank you for all the responses.
> > > > > > >
> > > > > > > My conclusion is that the paper can potentially be accepted, and if it does, the authors should embed this discussion into the paper.

---

> > > > > > > > ### Author Response · Authors · 2023-08-21
> > > > > > > >
> > > > > > > > We appreciate SSjS's many constructive comments.
> > > > > > > >
> > > > > > > > We will include the discussion of concave-1D and "monotonicity" in the final version. This will clarify the importance of our methods and provide clear avenues for further improvement.
> > > > > > > >
> > > > > > > > Once again, thank you for your insightful comments.

---

### Official Review · Reviewer_aq8D · 2023-07-05

**Soundness:** 2 fair
**Presentation:** 3 good
**Contribution:** 2 fair
**Rating:** 5
**Confidence:** 3

**Summary:**

The paper presents two improvements of the algorithm PLBF (Partitioned Learned Bloom Filter), called fast PLBF and fast PLBF++. PLBF learns the distribution structure and uses it to minimize the memory allocation. An ML model (LightGBM is used in this paper) is trained to predict a score between [0,1] indicating set membership probability. The score space is divided into N segments and then grouped into k<N regions that will each use a backup Bloom filter. The grouping of N segments into k region is formulated as an optimization problem and solved by dynamic programming, with time complexity O(N^3 k) for PLBF. The new method fast PLBF carefully refines the dynamic programming steps (by avoiding some redundant computation) and builds the same data structure but in O(N^2 k). The fast PLBF++ can construct a slightly different data structure, but faster; under some conditions, fastPLBF++ also constructs the same data structure. Experimental evaluation shows the performance of the two schemes.

**Strengths:**

Learned Bloom filters are a useful data structure. Improving the construction time for PLBF allows for the construction of filters for larger values of N, which improves their performance. The experimental evaluation is good.

**Weaknesses:**

It is not clearly explained what computations are redundant in PLBF and how they are avoided by fast PLBF.

**Questions:**

I would like to see more space devoted in the paper to the explanation of fast PLBF compared to PLBF. Can you explain in more detail, or even intuitively, what computation is redundant in PLBF and how it is avoided by the new scheme? All of it is deferred to the appendix, but even there it is not very clear. This is one of the main points of the paper and I think more justification should be included in the paper itself, not in the appendix.

---

> ### Author Rebuttal · Authors · 2023-08-09
>
> We appreciate the reviewer's insightful comments. We are happy to provide additional clarification on the unclear points raised.
>
> > It is not clearly explained what computations are redundant in PLBF and how they are avoided by fast PLBF.
>
> > Can you explain in more detail, or even intuitively, what computation is redundant in PLBF and how it is avoided by the new scheme?
>
> > This is one of the main points of the paper and I think more justification should be included in the paper itself, not in the appendix.
>
> Let us explain our approach intuitively. While PLBF builds many DP tables, our fast PLBF builds only one DP table and reuses it many times. Specifically, by exploiting the fact that the largest DP table $\mathrm{DP}^N_\mathrm{KL}$ contains other smaller DP tables, fast PLBF builds only $\mathrm{DP}^N_\mathrm{KL}$ and reuses it. This is a very simple speedup method, but it only becomes apparent by reorganizing the optimization problem and solution (which was not well organized in the original PLBF paper) and discovering that $j$ plays no role in a $\mathrm{DP}^j_\mathrm{KL}$ computation.
>
> Let us further elaborate on the optimization problem and how PLBF and fast PLBF solve it. To construct PLBFs, we need an optimal $\mathbf{t}$ and $\mathbf{f}$, where $\mathbf{t}$ are the thresholds for partitioning the score space into several regions and $\mathbf{f}$ are the false positive rates for each region. To find the optimal values, we must iterate over all the candidates of the “rightmost region” and finds the optimal $\mathbf{t}$ and $\mathbf{f}$ for each candidate (see Appendix C of the original PLBF paper for details). Since there are $N-k+1$ candidates of the rightmost region, there are $N-k+1$ “problems” to solve. PLBF solves $N-k+1$ problems separately, i.e., it computes the $N-k+1$ DP tables separately. Fast PLBF, on the other hand, computes only one DP table and leverages it repeatedly to solve $N-k+1$ problems. We mathematically showed that the results of PLBF and fast PLBF are identical.
>
> We will add this discussion to the final paper and detail the description of fast PLBF.

---

### Official Review · Reviewer_cwdf · 2023-07-14

**Soundness:** 4 excellent
**Presentation:** 3 good
**Contribution:** 3 good
**Rating:** 7
**Confidence:** 3

**Summary:**

This paper improves upon the Partitioned Learned Bloom Filter (PLBF), which is an essentially optimal learned Bloom filter introduced in ICML 2021. Construction time of PLBF is $O(N^3k)$, where $N > k$ are hyper-parameters. The authors show how to construct the same filter much more quickly; their Fast Partitioned Learned Bloom Filter (FPLBF) produces the same filter as PLBF and takes only $O(N^2k)$ time to construct. They also propose the fast PLBF++ algorithm that runs in even quicker $O(Nk\log N + Nk^2)$ time. It outputs the same filter under natural assumptions on the input and is essentially just as good experimentally. Extensive experimental evaluation with real world datasets and ablation studies support the claims and demonstrate significant practical speedups.

**Strengths:**

1) Bloom filters are a ubiquitous approximate set membership data structure. Their combination with learned predictors has been an active and fruitful direction of research in the past years. This paper is another significant step.
2) The authors are right (and the first to discover to the best of my knowledge) spot the redundancy in the PLBF paper and save an $O(N)$ factor.
3) Fast PLBF++ is a good connection with the monotone matrix max problem. The authors prove that under natural assumptions it provides the same output even quicker. It works well with real data even if the assumptions are not 100% satisfied.
4) Detailed experimental evaluation, especially kudos for the ablation in Section 5.3.

**Weaknesses:**

1) The PLBF authors were rather sloppy by rebuilding the same dynamic programming table from scratch when all they needed was adding another ''row'' to it. In fact looking at eq (2) in the paper it's quite obvious that superscript $j$ plays no role in $\mbox{DP}^j$ and a single DP table suffices. Nevertheless, credit is due for spotting this.
2) Practical runtime gains are potentially overstated as they depend on the choice of hyper-parameters $N$ and $k$. The paper shows runtime improvements for $N=1000$ and $k=5$, these hyperparameters are copied from the PLBF ICML 2021 paper, where they appeared without much justification. The (excellent) ablation experiments suggests that much smaller $N$ and much larger $k$ are optimal, which would decrease the runtime gaps in practice.

**Questions:**

Figure 12 shows that $N=50$ or $N=100$ max are sufficient for minimizing the false positive rate (for a given memory budget) for $k=5$. Figure 13 indicates that $k=100$ or even higher is required for minimizing the false positive rate for $N=1000$. Could you please find the $(N,k)$ combination that approximately minimizes the false positive rate and then measure and show the construction times for that? Could you please also zoom into the FPR plots around their minima without using log-scale on the x-axis (perhaps in the appendix) as I could not determine the location of the FPR minima in terms of $N$ even when I viewed the enlarged pdf on my screen,

**Limitations:**

Yes, it's adequate.

---

> ### Author Rebuttal · Authors · 2023-08-09
>
> We thank the reviewer for their positive evaluation of our paper. Let us answer the questions.
>
> > Could you please find the (N,k) combination that approximately minimizes the false positive rate and then measure and show the construction times for that?
>
> In Appendix E, we performed an ablation study on the hyperparameters N and k. In response to this comment, we performed an additional, more comprehensive ablation study.
>
> First, we performed a coarse grid search with N=10,20,50,100,200,500,1000 and k=3,5,7,10,20,50,100,200,500,1000 (note that k<=N). The memory usage of the backup Bloom filters was fixed at 500 Kb. The results are summarized as follows.
>
> In the case of URL dataset,
> - for PLBF: the FPR was minimal (1.25%) when (N, k)=(1000, 500).
>   - the construction time is 7.5 hours.
> - for fast PLBF: the FPR was minimal (1.25%) when (N, k)=(1000, 500).
>   - the construction time is 95.0 seconds (284 times faster than original PLBF).
> - for fast PLBF++: the FPR was minimal (1.26%) when (N, k)=(1000, 500).
>   - the construction time is 13.1 seconds (2064 times faster than original PLBF).
>
> In the case of Ember dataset,
> - for PLBF: the FPR was minimal (0.829%) when (N, k)=(500, 50).
>   - the construction time is 366 seconds.
> - for fast PLBF: the FPR was minimal (0.829%) when (N, k)=(500, 50).
>   - the construction time is 7.98 seconds (45 times faster than original PLBF).
> - for fast PLBF++: the FPR was minimal (0.857%) when (N, k)=(200, 50).
>   - the construction time is 5.67 seconds (64 times faster than original PLBF).
>
> These results coarsely give the (N, k) combination that minimizes the FPR. They also demonstrate that for the coarse optimal (N, k) settings, the original PLBF requires a substantial construction time, ranging from several minutes to several hours. On the other hand, our two proposed methods take only a few seconds to tens of seconds to build.
>
> > Could you please also zoom into the FPR plots around their minima without using log-scale on the x-axis (perhaps in the appendix) as I could not determine the location of the FPR minima in terms of N even when I viewed the enlarged pdf on my screen.
>
> Next, we performed a more detailed grid search of the fast PLBF accuracy around (N, k)=(1000, 500) for the URL dataset and (N, k)=(500, 50) for the Ember dataset. Specifically, N=500,750,1000,1250,1500 and k=200,400,500,600,800,1000,1250,1500 for the URL dataset and N=200,400,500,600,800,1000 and k=20,40,50,60,80,100 for the Ember dataset (again, note that k<=N). The memory usage of the backup Bloom filters was fixed at 500 Kb.
>
> In the case of URL dataset,
> - the FPR was minimal (1.23%) when (N, k)=(1500, 800).
>
> In the case of Ember dataset,
> - the FPR was minimal (0.829%) when (N, k)=(500, 50).
>
> This comprehensive ablation study and its appropriate visualization will be added to the final version.

---

> > ### Comment · Reviewer_cwdf · 2023-08-21
> >
> > Thanks for the additional experiments.
> >
> > If you read it in time: what were the speedups in the end with the optimal (N, k) parameters?
> >
> > I also appreciate Reviewer SSjS's insightful comments and promptly connecting these impactful practical results with theory of accelerated dynamic programming.

---

> > > ### Author Response · Authors · 2023-08-21
> > >
> > > We appreciate cwdf’s question and are happy to answer it.
> > >
> > > We are sorry, but due to lack of time, we cannot give a full experimental answer as to what the speedups were in the end with the optimal $(N, k)$. Although our estimates are included, the degree of speedup with the optimal $(N, k)$ can be summarized as follows:
> > >
> > > For the case of the URL dataset (minimum FPR is achieved with $(N, k)=(1500, 800)$),
> > > - Fast PLBF is **estimated** to be about 380 times faster than PLBF.
> > > - Fast PLBF++ is **estimated** to be about 3500 times faster than PLBF.
> > >
> > > For the case of the Ember dataset (minimum FPR is achieved with $(N, k) = (500, 50)$),
> > > - Fast PLBF is 45 times faster than PLBF.
> > > - Fast PLBF++ is 62 times faster than PLBF.
> > >
> > > The estimation was done as follows: the construction time of PLBF is asymptotically proportional to $N^3$ and proportional to $k$. With $(N, k) = (1000, 500)$, PLBF took 7.5 hours to build. So, we estimate that it would take about 40 hours to build with $(N, k) = (1500, 800)$. Note that the discussion period ends 3 hours from now, so we cannot run this experiment. With $(N, k)=(1500, 800)$, Fast PLBF takes 376 seconds, and Fast PLBF++ takes 40.6 seconds (these are actual measurements, not estimates). Thus, Fast PLBF and Fast PLBF++ are estimated to be about 380 and 3500 times faster than PLBF, respectively.
> > >
> > > We will include the actual experimental results in the final version.

---

### Decision · Program_Chairs · 2023-09-21

**Decision:**

Accept (poster)

**Comment:**

Authors presents two improvements of PLBF (Partitioned Learned Bloom Filter), called fast PLBF and fast PLBF++.  The results are sound, interesting and generally appreciated by all the reviewers. There were some concerns. After the rebuttal and thorough discussions, reviewers unanimously agreed that the paper is above the bar for publications.